# Geometric Neural Diffusion Processes

**Emile Mathieu**[*]
University Of Cambridge

**Vincent Dutordoir**[*]
University Of Cambridge
Secondmind Labs

**Michael J. Hutchinson**[*]
University Of Oxford

**Valentin De Bortoli**
Center for Science of Data, ENS Ulm

**Yee Whye Teh**
University Of Oxford

**Richard E. Turner**
University Of Cambridge,
Microsoft Research

## Abstract

Denoising diffusion models have proven to be a flexible and effective paradigm for generative modelling. Their recent extension to infinite dimensional Euclidean spaces has allowed for the modelling of stochastic processes. However, many problems in the natural sciences incorporate symmetries and involve data living in non-Euclidean spaces. In this work, we extend the framework of diffusion models to incorporate a series of geometric priors in infinite-dimension modelling. We do so by a) constructing a noising process which admits, as limiting distribution, a geometric Gaussian process that transforms under the symmetry group of interest, and b) approximating the score with a neural network that is equivariant w.r.t. this group. We show that with these conditions, the generative functional model admits the same symmetry. We demonstrate scalability and capacity of the model, using a novel Langevin-based conditional sampler, to fit complex scalar and vector fields, with Euclidean and spherical codomain, on synthetic and real-world weather data.

## 1 Introduction

Traditional denoising diffusion models are defined on finite-dimension Euclidean spaces (Song and Ermon, 2019; Song et al., 2021; Ho et al., 2020; Dhariwal and Nichol, 2021). Extensions have recently been developed for more exotic distributions, such as those supported on Riemannian manifolds (De Bortoli et al., 2022; Huang et al., 2022; Lou and Ermon, 2023; Fishman et al., 2023), and on function spaces of the form $f : \mathbb{R}^n \to \mathbb{R}^d$ (Dutordoir et al., 2022; Kerrigan et al., 2022; Lim et al., 2023a; Pidstrigach et al., 2023; Franzese et al., 2023; Bond-Taylor and Willcocks, 2023) (i.e. stochastic processes). In this work, we extend diffusion models to further deal with distributions over functions that incorporate non-Euclidean geometry in two different ways. This investigation of geometry also naturally leads to the consideration of symmetries in these distributions, and as such we also present methods for incorporating these into diffusion models.

Firstly, we look at *tensor fields*. Tensor fields are geometric objects that assign to all points on some manifold a value that lives in some vector space $V$. Roughly speaking, these are functions of the form $f : \mathcal{M} \to V$. These objects are central to the study of physics as they form a generic mathematical framework for modelling natural phenomena. Common examples include the pressure of a fluid in motion as $f : \mathbb{R}^3 \to \mathbb{R}$, representing wind over the Earth's surface as $f : \mathcal{S}^2 \to \mathrm{T}\mathcal{S}^2$, where $\mathrm{T}\mathcal{S}^2$ is the *tangent-space* of the sphere, or modelling the stress in a deformed object as $f : \text{Object} \to \mathrm{T}\mathbb{R}^3 \otimes \mathrm{T}\mathbb{R}^3$, where $\otimes$ is the *tensor-product* of the tangent spaces. Given the inherent symmetry in the laws of nature, these tensor fields can transform in a way that preserves these symmetries. Any modelling of these laws may benefit from respecting these symmetries.

37th Conference on Neural Information Processing Systems (NeurIPS 2023).

---

[*]Equal contribution

Secondly, we look at fields with manifold codomain, and in particular, at functions of the form $f : \mathbb{R} \to \mathcal{M}$. The challenge in dealing with manifold-valued output, arises from the lack of vector-space structure. In applications, these functions typically appear when modelling processes indexed by time that take values on a manifold. Examples include tracking the eye of cyclones moving on the surface of the Earth, or modelling the joint angles of a robot as it performs tasks.

The lack of data or noisy measurements in the physical process of interest motivates a *probabilistic* treatment of such phenomena, in addition to its functional nature. Arguably the most important framework for modelling stochastic processes are Gaussian Processes (GPs) (Rasmussen, 2003), as they allow for exact or approximate posterior prediction (Titsias, 2009; Rahimi and Recht, 2007; Wilson et al., 2020). In particular, when choosing equivariant mean and kernel functions, GPs are invariant (i.e. stationary) (Holderrieth et al., 2021; Azangulov et al., 2022; Azangulov et al., 2023). Their limited modelling capacity and the difficulty in designing complex, problem-specific kernels motivates the development of neural processes (NPs) (Garnelo et al., 2018b), which learn to approximately model a conditional stochastic process directly from data. NPs have been extended to model translation invariant (scalar) processes (Gordon et al., 2020) and more generic $\mathrm{E}(n)$-invariant processes (Holderrieth et al., 2021). Yet, the Gaussian conditional assumption of standard NPs still limits their flexibility and prevents such models from fitting complex processes. Diffusion models provide a compelling alternative for significantly greater modelling flexibility. In this work, we develop geometric diffusion neural processes which incorporate geometrical prior knowledge into functional diffusion models.

Our contributions are three-fold: (a) We extend diffusion models to more generic function spaces (i.e. tensor fields, and functions $f : \mathcal{X} \to \mathcal{M}$) by defining a suitable noising process. (b) We incorporate group invariance of the distribution of the generative model by enforcing the covariance kernel and the score network to be group equivariant. (c) We propose a novel Langevin dynamics scheme for efficient conditional sampling.

## 2 Background

**Denoising diffusion models.** We briefly recall here the key concepts behind diffusion models on $\mathbb{R}^d$ and refer the readers to (Song et al., 2021) for a more detailed introduction. We consider a forward *noising* process $(\mathbf{Y}_t)_{t\geq 0}$ defined by the following Stochastic Differential Equation (SDE)

$$\mathrm{d}\mathbf{Y}_t = -\tfrac{1}{2}\mathbf{Y}_t\mathrm{d}t + \mathrm{d}\mathbf{B}_t, \quad \mathbf{Y}_0 \sim p_0, \tag{1}$$

where $(\mathbf{B}_t)_{t\geq 0}$ is a $d$-dimensional Brownian motion and $p_0$ is the data distribution. The process $(\mathbf{Y}_t)_{t\geq 0}$ is simply an Ornstein–Ulhenbeck (OU) process which converges with geometric rate to $\mathrm{N}(0, \mathrm{Id})$ (Durmus and Moulines, 2017). Under mild conditions on $p_0$, the time-reversed process $(\bar{\mathbf{Y}}_t)_{t\geq 0} = (\mathbf{Y}_{T-t})_{t\in[0,T]}$ also satisfies an SDE (Cattiaux et al., 2021) given by

$$\mathrm{d}\bar{\mathbf{Y}}_t = \{\tfrac{1}{2}\bar{\mathbf{Y}}_t + \nabla \log p_{T-t}(\bar{\mathbf{Y}}_t)\}\mathrm{d}t + \mathrm{d}\mathbf{B}_t, \quad \bar{\mathbf{Y}}_0 \sim p_T, \tag{2}$$

where $p_t$ denotes the density of $\mathbf{Y}_t$. Unfortunately we cannot sample exactly from (2) as $p_T$ and the scores $(\nabla \log p_t)_{t\in[0,T]}$ are unavailable. First, $p_T$ is substituted with the limiting distribution $\mathrm{N}(0, \mathrm{Id})$ as it converges towards it. Second, one can easily show (Kallenberg, 2021, Thm 5.1) that the score $\nabla \log p_t$ is the minimiser of $\ell_t(\mathbf{s}) = \mathbb{E}[\|\mathbf{s}(\mathbf{Y}_t) - \nabla_{y_t} \log p_{t|0}(\mathbf{Y}_t|\mathbf{Y}_0)\|^2]$ over functions $\mathbf{s} : [0,T] \times \mathbb{R}^d \to \mathbb{R}^d$ where the expectation is over the joint distribution of $\mathbf{Y}_0, \mathbf{Y}_t$, and as such can readily be approximated by a neural network $\mathbf{s}_\theta(t, y_t)$ by minimising this functional. Finally, a discretisation of (2) is performed (e.g. Euler–Maruyama) to obtain approximate samples of $p_0$.

**Steerable fields.** We now define steerable feature fields which are collections of tensor fields. We focus on the Euclidean group $G = \mathrm{E}(d)$, which elements $g$ admits a unique decomposition $g = uh$ where $h \in \mathrm{O}(d)$ is a $d \times d$ orthogonal matrix and $u \in \mathrm{T}(d)$ is a translation which can be identified as an element of $\mathbb{R}^d$; for a vector $x \in \mathbb{R}^d$, $g \cdot x = hx + u$ denotes the action of $g$ on $x$, with $h$ acting from the left on $x$ by matrix multiplication. This special case simplifies the presentation, but can be extended to the general case is discussed in App. C.1.

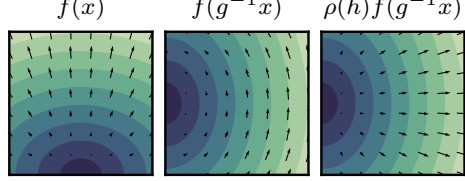

$f(x) \qquad f(g^{-1}x) \qquad \rho(h)f(g^{-1}x)$

Figure 1: Illustration of a vector field $f : \mathbb{R}^2 \to \mathbb{R}^2$ with representation $\rho(h) = h$ being steered by a group element $h = 90° \in \mathrm{O}(2) \subset \mathrm{E}(2)$.

We are interested in learning a probabilistic model over functions of the form $f : \mathcal{X} \to \mathbb{R}^d$ such that a group $G$ acts on $\mathcal{X}$ and $\mathbb{R}^d$. We call a feature field a tuple $(f, \rho)$ with $f : \mathcal{X} \to \mathbb{R}^d$ a mapping between input $x \in \mathcal{X}$ to some feature $f(x)$ with associated representation $\rho : G \to \mathrm{GL}(\mathbb{R}^d)$ (Scott and Serre, 1996). This feature field is said to be $G$-steerable if it is transformed for all $x \in \mathcal{X}, g \in G$ as $g \cdot f(x) = \rho(g)f(g^{-1} \cdot x)$. In this setting, the action of $\mathrm{E}(d) = \mathrm{T}(d) \rtimes \mathrm{O}(d)$ on the feature field $f$ yields $g \cdot f(x) = (uh) \cdot f(x) \triangleq \rho(h)f\left(h^{-1}(x - u)\right)$. Typical examples of feature fields include scalar fields with $\rho_{\mathrm{triv}}(g) \triangleq 1$ transforming as $g \cdot f(x) = f\left(g^{-1}x\right)$ such as temperature fields, and vectors or potential fields with $\rho_{\mathrm{Id}}(g) \triangleq h$ transforming as $g \cdot f(x) = hf\left(g^{-1}x\right)$ as illustrated in Fig. 1, such as wind or force fields.

For many natural phenomena, a priori we do not want to express a preference for a particular conformation of the feature field and thus want a prior $p$ to place the same density on all the transformed fields $\mu(g \cdot f) = \mu(f), \ \forall g \in G$. Leveraging this symmetry can drastically reduce the amount of data required to learn from and reduce training time.

## 3 Geometric neural diffusion processes: GEOMNDPS

### 3.1 Continuous diffusion on function spaces

We construct a diffusion model on functions $f : \mathcal{X} \to \mathcal{Y}$, with $\mathcal{Y} = \mathbb{R}^d$, by defining a diffusion model for every finite set of marginals. Most prior works on infinite-dimensional diffusions consider a noising process on the space of functions (Kerrigan et al., 2022; Pidstrigach et al., 2023; Lim et al., 2023b). In theory, this allows the model to define a consistent distribution over all the finite marginals of the process being modelled. In practice, however, only finite marginals can be modelled on a computer and the score function needs to be approximated, and at this step lose consistency over the marginals. The only work to stay fully consistent in implementation is Phillips et al. (2022), at the cost of limiting functions that can be modelled to a finite-dimensional subspace. With this in mind, we eschew the technically laborious process of defining diffusions over the infinite-dimension space and work solely on the finite marginals following Dutordoir et al. (2022). We find that in practice consistency can be well learned from data see Sec. 5, and this allows for more flexible choices of score network architecture and easier training.

**Noising process.** We assume we are given a data process $(\mathbf{Y}_0(x))_{x \in \mathcal{X}}$. Given any $x = (x^1, \ldots, x^n) \in \mathcal{X}^n$, we consider the following forward *noising* process $(\mathbf{Y}_t(x))_{t \geq 0} \triangleq (\mathbf{Y}_t(x^1), \ldots, \mathbf{Y}_t(x^n))_{t \geq 0} = (\mathbf{Y}_t^1, \ldots, \mathbf{Y}_t^n)_{t \geq 0} \in \mathcal{Y}^n$ defined by the following SDE

$$d\mathbf{Y}_t(x) = \tfrac{1}{2}\left\{ m(x) - \mathbf{Y}_t(x) \right\} \beta_t dt + \beta_t^{1/2} \mathrm{K}(x, x)^{1/2} d\mathbf{B}_t, \tag{3}$$

where $\mathrm{K}(x, x)_{i,j} = k(x^i, x^j)$ with $k : \mathcal{X} \times \mathcal{X} \to \mathbb{R}$ a kernel and $m : \mathcal{X} \to \mathcal{Y}$. The process $(\mathbf{Y}_t(x))_{t \geq 0}$ is a multivariate Ornstein–Uhlenbeck process—with drift $b(t, x, \mathbf{Y}_t(x)) = m(x) - \mathbf{Y}_t(x)$ and diffusion coefficient $\sigma(t, x, \mathbf{Y}_t(x)) = \mathrm{K}(x, x)$—which converges with geometric rate to $\mathrm{N}(m(x), \mathrm{K}(x, x))$. Using Phillips et al. (2022), it can be shown that this convergence extends to the *process* $(\mathbf{Y}_t)_{t \geq 0}$ which converges to the Gaussian Process with mean $m$ and kernel $k$, denoted $\mathbf{Y}_\infty$.

In the specific instance where $k(x, x') = \delta_x(x')$, then the limiting process $\mathbf{Y}_\infty$ is simply Gaussian *white noise*, whilst other choices such as the squared-exponential or Matérn kernel would lead to the associated Gaussian limiting process $\mathbf{Y}_\infty$. Note that the *white noise* setting is not covered by the existing theory of functional diffusion models, as a Hilbert space and a square integral kernel are required, see Kerrigan et al. (2022) for instance.

**Denoising process.** Under mild conditions over $\mathbf{Y}_0$[2], the time-reversal process $(\bar{\mathbf{Y}}_t(x))_{t \geq 0}$ satisfies

$$d\bar{\mathbf{Y}}_t(x) = \{-\tfrac{1}{2}(m(x) - \bar{\mathbf{Y}}_t(x)) + \mathrm{K}(x, x)\nabla \log p_{T-t}(\bar{\mathbf{Y}}_t(x))\}\beta_{T-t}dt + \beta_{T-t}^{1/2}\mathrm{K}(x, x)^{1/2}d\mathbf{B}_t, \tag{4}$$

with $\bar{\mathbf{Y}}_0 \sim \mathrm{GP}(m, k)$ and $p_t$ the density of $\mathbf{Y}_t(x)$ w.r.t. Lebesgue. In practice, the $\log p_{T-t}$ term–known as the Stein score, is not tractable and must be approximated by a neural network. We then consider the generative stochastic process model defined by first sampling $\bar{\mathbf{Y}}_0 \sim \mathrm{GP}(m, k)$ and then simulating the reverse diffusion (4) (e.g. via Euler-Maruyama discretisation).

---

[2]Haussmann and Pardoux (1986) assumes Lipschitz continuity of the drift and volatility matrix and Cattiaux et al. (2021, Thm 4.9) which only assume a finite entropy condition on the space of processes.

**Manifold valued outputs.** So far we have defined our generative model with $\mathcal{Y} = \mathbb{R}^d$, we can readily extend the methodology to manifold-valued functional models using *Riemannian* diffusion models such as De Bortoli et al. (2022) and Huang et al. (2022), see App. C. One of the main notable difference is that in the case where $\mathcal{Y}$ is a *compact* manifold, we replace the Ornstein-Uhlenbeck process by a Brownian motion which targets the uniform distribution.

**Training.** As the reverse SDE (4) involves the preconditioned score $\mathrm{K}(x,x)\nabla \log p_t$, we directly approximate it with a neural network $(\mathrm{Ks})_\theta : [0,T] \times \mathcal{X}^n \times \mathcal{Y}^n \to \mathrm{T}\mathcal{Y}^n$, where $\mathrm{T}\mathcal{Y}$ is the tangent bundle of $\mathcal{Y}$, see App. C. The conditional score of the noising process (3) is given by

$$\nabla_{\mathbf{Y}_t} \log p_t(\mathbf{Y}_t(x)|\mathbf{Y}_0(x)) = -\Sigma_{t|0}^{-1}(\mathbf{Y}_t(x) - m_{t|0}) = -\sigma_{t|0}^{-1}\mathrm{K}(x,x)^{-1/2}\varepsilon, \tag{5}$$

since $\mathbf{Y}_t = m_{t|0} + \Sigma_{t|0}^{1/2}\varepsilon$ with $\varepsilon \sim \mathrm{N}(0, \mathrm{Id})$, and $\Sigma_{t|0} = \sigma_{t|0}^2 \mathrm{K}$ with $\sigma_{t|0} = (1 - e^{-\int_0^t \beta(s)\mathrm{d}s})^{1/2}$, see App. B.1. We learn the preconditioned score $(\mathrm{Ks})_\theta$ by minimising the following denoising score matching (DSM) loss (Vincent et al., 2010) weighted by $\Lambda(t) = \sigma_{t|0}^2 \mathrm{K}^\top \mathrm{K}$

$$\mathcal{L}(\theta; \Lambda(t)) = \mathbb{E}[\|s_\theta(t, \mathbf{Y}_t) - \nabla \log p_t(\mathbf{Y}_t|\mathbf{Y}_0)\|^2_{\Lambda(t)}] = \mathbb{E}[\|\sigma_{t|0} \cdot (\mathrm{K}s)_\theta(t, \mathbf{Y}_t) + \mathrm{K}^{1/2}\varepsilon\|^2_2], \tag{6}$$

where $\|x\|^2_\Lambda = x^\top \Lambda x$. Note that when targeting a unit-variance white noise, then $\mathrm{K} = \mathrm{Id}$ and the loss (6) reverts to the DSM loss with weighting $\lambda(t) = 1/\sigma_{t|0}^2$ (Song et al., 2021). In App. B.2, we explore several preconditioning terms and associated weighting $\Lambda(t)$. Overall, we found the preconditioned score $\mathrm{K}\nabla \log p_t$ parameterisation, in combination with the $\ell_2$ loss, to perform best, as shown by the ablation study in App. F.1.3.

### 3.2 Invariant neural diffusion processes

In this section, we show how we can incorporate geometrical constraints into the functional diffusion model introduced in the previous Sec. 3.1. In particular, given a group $G$, we aim to build a generative model over steerable feature fields as defined in Sec. 2.

**Invariant process.** A stochastic process $f \sim \mu$ is said to be $G-$invariant if $\mu(g \cdot \mathrm{A}) = \mu(\mathrm{A})$ for any $g \in G$, with $\mu \in \mathcal{P}(\mathrm{C}(\mathcal{X}, \mathcal{Y}))$, where $\mathcal{P}$ is the space of probability measure on the space of continuous functions and $\mathrm{A} \subset \mathrm{C}(\mathcal{X}, \mathcal{Y})$ measurable. From a sample perspective, this means that with input-output pairs $\mathcal{C} = \{(x^i, y^i)\}_{i=1}^n$, and denoting the action of $G$ on this set as $g \cdot \mathcal{C} \triangleq \{(g \cdot x^i, \rho(g)y^i)\}_{i=1}^n$, $f \sim \mu$ is $G-$invariant if and only if $g \cdot \mathcal{C}$ has the same distribution as $\mathcal{C}$. In what follows, we aim to derive sufficient conditions on the model introduced in Sec. 3 so that it satisfies this $G$-invariance property. First, we recall such a necessary and sufficient condition for Gaussian processes.

**Proposition 3.1.** *Invariant (stationary) Gaussian process (Holderrieth et al., 2021). We have that a Gaussian process* $\mathrm{GP}(m, k)$ *is $G$-invariant if and only if its mean $m$ and covariance $k$ are suitably $G$-equivariant—that is, for all $x, x' \in \mathcal{X}, g \in G$*

$$m(g \cdot x) = \rho(g)m(x) \ \text{ and } \ k(g \cdot x, g \cdot x') = \rho(g)k(x, x')\rho(g)^\top. \tag{7}$$

Trivial examples of $\mathrm{E}(n)$-equivariant kernels include diagonal kernels $k = k_0 \mathrm{Id}$ with $k_0$ invariant (Holderrieth et al., 2021), but see App. F.2 for non trivial instances introduced by Macêdo and Castro (2010). Building on Prop. 3.1, we then state that our introduced neural diffusion process is also invariant if we additionally assume the score network to be $G$-equivariant.

**Proposition 3.2.** *Invariant neural diffusion process (Yim et al., 2023, Prop 3.6). We denote by $(\bar{\mathbf{Y}}_t(x))_{x \in \mathcal{X}, t \in [0,T]}$ the process induced by the time-reversal SDE (4) where the score is approximated by a score network $s_\theta : [0,T] \times \mathcal{X}^n \times \mathcal{Y}^n \to \mathrm{T}\mathcal{Y}^n$, and the limiting process is given by $\mathcal{L}(\bar{\mathbf{Y}}_0) = \mathrm{GP}(m, k)$. Assuming $m$ and $k$ are respectively $G$-equivariant per Prop. 3.1, if we additionally have that the score network is $G$-equivariant vector field, i.e. $s_\theta(t, g \cdot x, \rho(g)y) = \rho(g)s_\theta(t, x, y)$ for all $x \in \mathcal{X}, g \in G$, then for any $t \in [0,T]$ the process $(\bar{\mathbf{Y}}_t(x))_{x \in \mathcal{X}}$ is $G$-invariant.*

This result can be proved in two ways, from the probability flow ODE perspective or directly in terms of SDE via Fokker-Planck, see App. D.2. In particular, when modelling an invariant scalar data process $(\mathbf{Y}_0(x))_{x \in \mathcal{X}}$ such as a temperature field, we need the score network to admit the invariance constraint $s_\theta(t, g \cdot x, y) = s_\theta(t, x, y)$.

**Equivariant conditional process.** Often precedence is given to modelling the predictive process

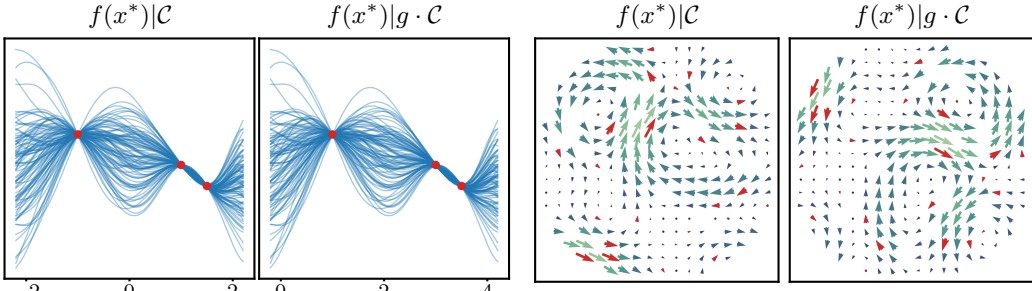

Figure 2: Samples from equivariant neural diffusion processes conditioned on context set $\mathcal{C}$ (in red) and evaluated on a regular grid $x^*$ for scalar (*Left*) and 2D vector (*Right*) fields. Same model is then conditioned on transformed context $g \cdot \mathcal{C}$, with group element $g$ being a translation of length 2 (*Left*) or a $90°$ rotation (*Right*).

given a set of observations $\mathcal{C} = \{(x^c, y^c)\}_{c \in C}$. In this context, the conditional process (Pollard, 2002, p.117) inherits the symmetry of the prior process in the following sense. A stochastic process with distribution $\mu$ given a context $\mathcal{C}$ is said to be conditionally $G-$equivariant if the conditional satisfies $\mu(\mathsf{A}|g \cdot \mathcal{C}) = \mu(g \cdot \mathsf{A}|\mathcal{C})$ for any $g \in G$ and $\mathsf{A} \in \mathrm{C}(\mathcal{X}, \mathcal{Y})$ measurable, as illustrated in Fig. 2.

**Proposition 3.3.** *Equivariant conditional process.* *Assume a stochastic process $f \sim \mu$ is $G-$invariant. Then the conditional process $f|\mathcal{C}$ given a set of observations $\mathcal{C}$ is $G$-equivariant.*

Originally stated in Holderrieth et al. (2021) in the case where the process is over functions of the form $f : \mathbb{R}^n \to \mathbb{R}^d$ and marginals with density w.r.t. Lebesgue, we prove Prop. 3.3 for stochastic processes over generic fields on manifolds in terms only of the measure of the process (App. D.3).

### 3.3 Conditional sampling

There exist several methods to perform conditional sampling in diffusion models such as: replacement sampling, amortisation and conditional guidance, which we discuss in App. E.1. Here we propose a new method for sampling from exact conditional distributions of NDPs using only the score network for the joint distribution. Using the fact that the conditional score can be written as $\nabla_x \log p(x|y) = \nabla_x \log p(x, y) - \nabla_x \log p(y) = \nabla_x \log p(x, y)$ we can therefore, for any point in the diffusion time, conditionally sample using Langevin dynamics, following the SDE $\mathrm{d}\mathbf{Y}_s = \frac{1}{2}\mathrm{K}\nabla \log p_{T-t}(\mathbf{Y}_s)\mathrm{d}s +$

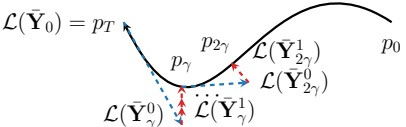

Figure 3: Illustration of Langevin corrected conditional sampling. The black line represents the noising process dynamics $(p_t)_{t \in [0,T]}$. The time reversal (i.e. predictor) step, is combined with a Langevin corrector step projecting back onto the dynamics.

$\sqrt{\mathrm{K}}\mathrm{d}\mathbf{B}_s$, by only applying the diffusion to the variables of interest and holding the others fixed. While we could sample directly at the end time this proves difficult in practice. Similar to the motivation of Song and Ermon (2019), we sample along the reverse diffusion, taking a number of conditional Langevin steps at each time. In addition, we apply the forward noising SDE to the conditioning points at each step, as this puts the combined context and sampling set in a region that the score function will be well learned in training. Our procedure is illustrated in Alg. 1. In App. E.1 we draw links with REPAINT of Lugmayr et al. (2022).

### 3.4 Likelihood evaluation

Similarly to Song et al. (2021), we can derive a deterministic ODE which has the same marginal density as the SDE (3), which is given by the 'probability flow' Ordinary Differential Equation (ODE), see App. B. Once the score network is learnt, we can thus use it in conjunction with an ODE solver to compute the likelihood of the model. A perhaps more interesting task is to evaluate the predictive posterior likelihood $p(y^*|x^*, \{x^i, y^i\}_{i \in C})$ given a context set $\{x^i, y^i\}_{i \in C}$. A simple approach is to simply rely on the conditional probability rule evaluate $p(y^*|x^*, \{x^i, y^i\}_{i \in C}) = p(y^*, \{y^i\}_{i \in C}|x^*, \{x^i\}_{i \in C})/p(\{y^i\}_{i \in C}|\{x^i\}_{i \in C})$. This can be done by solving two probability flow ODEs: one over the joint evaluation and context set, and another only over the context set.

# 4 Related work

**Gaussian processes and the neural processes family.** One important and powerful framework to construct distributions over functional spaces are Gaussian processes (Rasmussen, 2003). Yet, they are restricted in their modelling capacity and when using exact inference they scale poorly with the number of datapoints. These problems can be partially alleviated by using neural processes (Kim et al., 2019; Garnelo et al., 2018b; Garnelo et al., 2018a; Jha et al., 2022; Louizos et al., 2019; Singh et al., 2019), although they also assume a Gaussian likelihood. Recently introduced autoregressive NPs (Bruinsma et al., 2023) alleviate this limitation, but they are disadvantaged by the fact that variables early in the auto-regressive generation only have simple distributions (typically Gaussian). Finally, (Dupont et al., 2022) model weights of implicit neural representation using diffusion models.

**Stationary stochastic processes.** The most popular Gaussian process kernels (e.g. squared exponential, Matérn) are stationary, that is, they are translation invariant. These lead to invariant Gaussian processes, whose samples when translated have the same distribution as the original ones. This idea can be extended to the entire isometry group of Euclidean spaces (Holderrieth et al., 2021), allowing for modelling higher order tensor fields, such as wind fields or incompressible fluid velocity (Macêdo and Castro, 2010). Later, Azangulov et al. (2022) and Azangulov et al. (2023) extended stationary kernels and Gaussian processes to a large class of non-Euclidean spaces, in particular all compact spaces, and symmetric non compact spaces. In the context of neural processes, (Gordon et al., 2020) introduced CONVCNP so as to encode translation equivariance into the predictive process. They do so by embedding the context into a translation equivariant functional representation which is then decoded with a convolutional neural network. Holderrieth et al. (2021) later extended this idea to construct neural processes that are additionally equivariant w.r.t. rotations or subgroup thereof.

**Spatial structure in diffusion models.** A variety of approaches have also been proposed to incorporate spatial correlation in the noising process of finite-dimensional diffusion models leveraging the multiscale structure of data (Jing et al., 2022; Guth et al., 2022; Ho et al., 2022; Saharia et al., 2021; Hoogeboom and Salimans, 2022; Rissanen et al., 2022). Our methodology can also be seen as a principled way to modify the forward dynamics in classical denoising diffusion models. Hence, our contribution can be understood in the light of recent advances in generative modelling on soft and cold denoising diffusion models (Daras et al., 2022; Bansal et al., 2022; Hoogeboom and Salimans, 2022). Several recent work explicitly introduced a covariance matrix in the Gaussian noise, either on a choice of kernel (Biloš et al., 2022), based on Discrete Fourier Transform of images (Voleti et al., 2022), or via empirical second order statistics (squared pairwise distances and the squared radius of gyration) for protein modelling (Ingraham et al., 2022). Alternatively, (Guth et al., 2022) introduced correlation on images leveraging a wavelet basis.

**Functional diffusion models.** Infinite dimensional diffusion models have been investigated in the Euclidean setting in (Kerrigan et al., 2022; Pidstrigach et al., 2023; Lim et al., 2023b; Bond-Taylor and Willcocks, 2023; Hagemann et al., 2023; Franzese et al., 2023; Dutordoir et al., 2022; Phillips et al., 2022). Most of these works are based on an extension of the diffusion models techniques (Song et al., 2021; Ho et al., 2020) to the infinite-dimensional space, leveraging tools from the Cameron-Martin theory such as the Feldman-Hájek theorem (Kerrigan et al., 2022; Pidstrigach et al., 2023) to define infinite-dimensional Gaussian measures and how they interact. We refer to (Da Prato and Zabczyk, 2014) for a thorough introduction to Stochastic Differential Equations in infinite dimension. (Phillips et al., 2022) consider another approach by defining countable diffusion processes in a basis. All these approaches amount to learn a diffusion model with spatial structure. Note that this induced correlation is necessary for the theory of infinite dimensional SDE (Da Prato and Zabczyk, 2014) to be applied but is not necessary to implement diffusion models (Dutordoir et al., 2022). Several approaches have been considered for conditional sampling. (Pidstrigach et al., 2023; Bond-Taylor and Willcocks, 2023) modify the reverse diffusion to introduce a guidance term, while (Dutordoir et al., 2022; Kerrigan et al., 2022) use the replacement method. Finally (Phillips et al., 2022) amortise the score function w.r.t. the conditioning context.

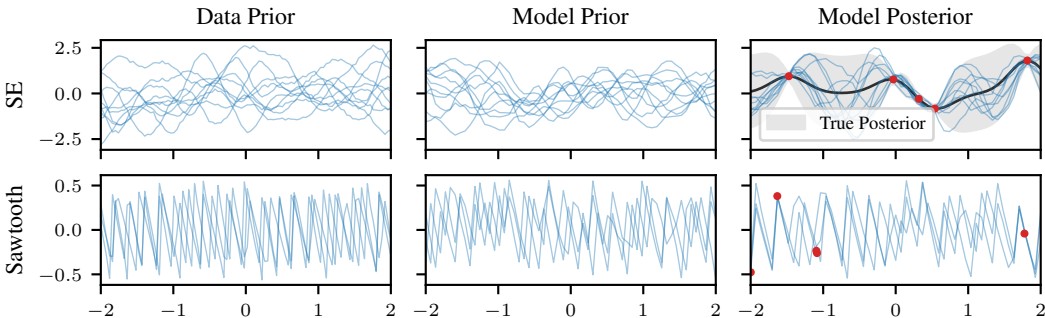

Figure 4: Prior and posterior samples (in blue) from the data process and the GeomNDP model, with context points in red and posterior mean in black.

# 5 Experimental results

## 5.1 1D regression over stationary scalar fields

We evaluate GEOMNDPS on several synthetic 1D regression datasets. We follow the same experimental setup as Bruinsma et al. (2020) which we detail in App. F.1. In short, it contains Gaussian (Squared Exponential (SE), MATÉRN($\frac{5}{2}$), WEAKLY PERIODIC) and non-Gaussian (SAWTOOTH and MIXTURE) sample paths, where MIXTURE is a combination of the other four datasets with equal weight. Fig. 9 shows samples for each of these dataset. The Gaussian datasets are corrupted with observation noise with variance $\sigma^2 = 0.05^2$. Table 1 reports the average log-likelihood $p(y^*|x^*, \mathcal{C})$ across 4096 test samples, where the context set size is uniformly sampled between 1 and 10 and the target has fixed size of 50. All inputs $x^c, x^*$ are chosen uniformly within their input domain which is [-2, 2] for the training data and 'interpolation' evaluation and [2, 6] for the 'generalisation' evaluation.

We compare the performance of GeomNDP to a GP with the true hyperparameters (when available), a (convolutional) Gaussian NP (Bruinsma et al., 2020), a convolutional NP (Gordon et al., 2020) and a vanilla attention-based NDP (Dutordoir et al., 2022) which we reformulated in the continuous diffusion process framework to allow for log-likelihood evaluations and thus a fair comparison—denoted NDP*. We enforce translation invariance in the score network for GeomNDP by subtracting the centre of mass from the input $x$, inducing stationary scalar fields.

On the GP datasets, GNP, CONVNPS and GeomNDP methods are able to fit the conditionals perfectly—matching the log-likelihood of the GP model. GNP's performance degrades on the non-Gaussian datasets as it is restricted by its conditional Gaussian assumption, whilst NDPs methods still performs well as illustrated on Fig. 4. In the bottom rows of Table 1, we assess the models ability to generalise outside of the training input range $x \in$ [-2, 2], and evaluate them on a translated grid where context and target points are sampled from [2, 6]. Only convolutional NPs (GNP and CONVNP) and T(1)−GEOMNDP are able to model stationary processes and therefore to perform as well as in the interpolation task. The NDP*, on the contrary, drastically fails at this task.

Table 1: Mean test log-likelihood (TLL) (↑) $\pm$ 1 standard error estimated over 4096 test samples are reported. Statistically significant best non-GP model is in **bold**. '*' stands for a TLL below −10. NP baselines from Bruinsma et al. (2020). T(1)−GeomNDP indicates our proposed method with a translation invariant score.

| | | SE | MATÉRN($\frac{5}{2}$) | WEAKLY PER. | SAWTOOTH | MIXTURE |
|---|---|---|---|---|---|---|
| **INTERPOLAT.** | GP (OPTIMUM) | $0.70\pm_{0.00}$ | $0.31\pm_{0.00}$ | $-0.32\pm_{0.00}$ | - | - |
| | T(1)−GEOMNDP | $\mathbf{0.72}\pm_{0.03}$ | $\mathbf{0.32}\pm_{0.03}$ | $\mathbf{-0.38}\pm_{0.03}$ | $\mathbf{3.39}\pm_{0.04}$ | $\mathbf{0.64}\pm_{0.08}$ |
| | NDP* | $\mathbf{0.71}\pm_{0.03}$ | $0.30\pm_{0.03}$ | $-0.37\pm_{0.03}$ | $\mathbf{3.39}\pm_{0.04}$ | $\mathbf{0.64}\pm_{0.08}$ |
| | GNP | $\mathbf{0.70}\pm_{0.01}$ | $\mathbf{0.30}\pm_{0.01}$ | $-0.47\pm_{0.01}$ | $0.42\pm_{0.01}$ | $0.10\pm_{0.02}$ |
| | CONVNP | $-0.46\pm_{0.01}$ | $-0.67\pm_{0.01}$ | $-1.02\pm_{0.01}$ | $1.20\pm_{0.01}$ | $-0.50\pm_{0.02}$ |
| **GENERALISAT.** | GP (OPTIMUM) | $0.70\pm_{0.00}$ | $0.31\pm_{0.00}$ | $-0.32\pm_{0.00}$ | | |
| | T(1)−GEOMNDP | $\mathbf{0.70}\pm_{0.02}$ | $\mathbf{0.31}\pm_{0.02}$ | $\mathbf{-0.38}\pm_{0.03}$ | $\mathbf{3.39}\pm_{0.03}$ | $\mathbf{0.62}\pm_{0.02}$ |
| | NDP* | * | * | * | * | * |
| | GNP | $\mathbf{0.69}\pm_{0.01}$ | $\mathbf{0.30}\pm_{0.01}$ | $-0.47\pm_{0.01}$ | $0.42\pm_{0.01}$ | $0.10\pm_{0.02}$ |
| | CONVNP | $-0.46\pm_{0.01}$ | $-0.67\pm_{0.01}$ | $-1.02\pm_{0.01}$ | $1.19\pm_{0.01}$ | $-0.53\pm_{0.02}$ |

| MODEL | SE | CURL-FREE | DIV-FREE |
|---|---|---|---|
| GP | $0.56_{\pm 0.00}$ | $0.66_{\pm 0.00}$ | $0.66_{\pm 0.00}$ |
| NDP* | $0.55_{\pm 0.00}$ | $0.62_{\pm 0.01}$ | $0.62_{\pm 0.01}$ |
| E(2)-GEOMNDP | $\mathbf{0.56_{\pm 0.01}}$ | $\mathbf{0.65_{\pm 0.01}}$ | $\mathbf{0.66_{\pm 0.01}}$ |
| GP (DIAG.) | $-1.56_{\pm 0.00}$ | $-1.47_{\pm 0.00}$ | $-1.47_{\pm 0.00}$ |
| T(2)-CONVCNP | $-1.71_{\pm 0.01}$ | $-1.77_{\pm 0.01}$ | $-1.76_{\pm 0.00}$ |
| E(2)-STEERCNP | $-1.61_{\pm 0.00}$ | $-1.57_{\pm 0.00}$ | $-1.57_{\pm 0.01}$ |

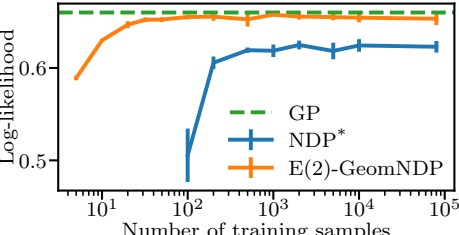

Figure 5: Quantitative results for experiments on GP vector fields. Mean predictive log-likelihood ($\uparrow$) and confidence interval estimated over 5 random seeds. *Left*: Comparison with neural processes. Statistically significant results are in **bold**. *Right*: Ablation study when varying the number of training data samples.

**Non white kernels for limiting process.** The NDP methods in the above experiment target the white kernel $\mathbb{1}(x = x')$ in the limiting process. In App. F.1.3, we explore different choices for the limiting kernel, such as SE and periodic kernels with short and long lengthscales, along with several score parameterisations, see App. B.3 for a description of these. We observe that although choosing such kernels gives a head start to the training, it eventually yield slightly worse performance. We attribute this to the additional complexity of learning a non-diagonal covariance. Finally, across all datasets and limiting kernels, we found the preconditioned score $K\nabla \log p_t$ to result in the best performance.

**Conditional sampling ablation.** We employ the SE dataset to investigate various configurations of the conditional sampler as we have access to the ground truth conditional distribution through the GP posterior. In Fig. 11 we compute the Kullback-Leibler divergence between the samples generated by GEOMNDP and the actual conditional distribution across different conditional sampling settings. Our results demonstrate the importance of performing multiple Langevin dynamics steps during the conditional sampling process. Additionally, we observe that the choice of noising scheme for the context values $y_c$ has relatively less impact on the overall outcome.

### 5.2 Regression over Gaussian process vector fields

We now focus our attention to modelling equivariant vector fields. For this, we create datasets using samples from a two-dimensional zero-mean GP with one of the following $E(2)$-equivariant kernels: a diagonal Squared-Exponential (SE) kernel, a zero curl (CURL-FREE) kernel and a zero divergence (DIV-FREE) kernel, as described in App. D.1.

We equip our model, GeomNDP, with a $E(2)$-equivariant score architecture, based on steerable CNNs (Thomas et al., 2018; Weiler and Cesa, 2021). We compare to NDP* with a non-equivariant attention-based network (Dutordoir et al., 2022). We also evaluate two neural processes, a translation-equivariant CONVCNP (Gordon et al., 2020) and a $C4 \ltimes \mathbb{R}^2 \subset E(2)$-equivariant STEERCNP (Holderrieth et al., 2021). We also report the performance of the data-generating GP, and the same GP but with diagonal posterior covariance GP (DIAG.). We measure the predictive log-likelihood of the data process samples under the model on a held-out test dataset.

We observe in Fig. 5 (Left), that the CNPs performance is limited by their diagonal predictive covariance assumption, and as such cannot do better than the GP (DIAG.). We also see that although NDP* is able to fit well GP posteriors, it does not reach the maximum log-likelihood value attained by the data GP, in contrast to its equivariant counterpart GEOMNDP. To further explore gains brought by the built-in equivariance, we explore the data-efficiency in Fig. 5 (Right), and notice that E(2)-GEOMNDP requires few data samples to fit the data process, since effectively the dimension of the (quotiented) state space is dramatically reduced.

### 5.3 Global tropical cyclone trajectory prediction

Finally, we assess our model on a task where the domain of the stochastic process is a non-Euclidean manifold. We model the trajectories of cyclones over the earth, modelled as sample paths of the form $\mathbb{R} \to \mathcal{S}^2$ coming from a stochastic process. The data is drawn from the International Best Track Archive for Climate Stewardship (IBTrACS) Project, Version 4 ((Knapp et al., 2010; Knapp et al., 2018)) and preprocessed as per App. F.3, where details on the implementation of the score function, the ODE/SDE solvers used for the sampling, and baseline methods can be found.

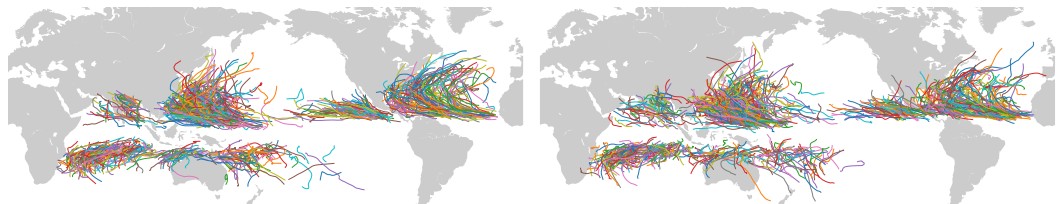

Figure 6: *Left:* 1000 samples from the training data. *Right:* 1000 samples from the trained model.

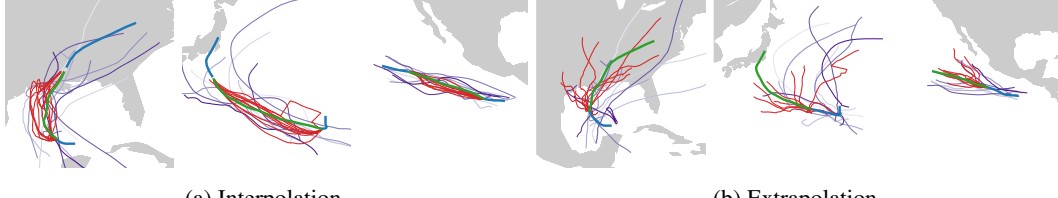

(a) Interpolation                                      (b) Extrapolation

Figure 7: *Top:* Examples of conditional trajectories sampled from the GeomNDP model. *Blue:* Conditioned sections of the trajectory. *Green:* The actual trajectory of the cyclone. *Red:* conditional samples from the model. *Purple:* closest matching trajectories in the dataset to the conditioning data.

Fig. 6 shows some cyclone trajectories samples from the data process and from a trained GEOMNDP model. We also demonstrate how such trajectories can be interpolated or extrapolated using the conditional sampling method detailed in Sec. 3.3. Such conditional sample paths are shown in Fig. 7. Additionally, we report in Table 2 the likelihood and MSE for a series of methods. The interpolation task involves conditioning on the first and last 20% of the cyclone trajectory and predicting intermediary positions. The extrapolation task involves conditioning on the first 40% of trajectories and predicting future positions. We see that the GPs (modelled as $f : \mathbb{R} \to \mathbb{R}^2$, one on latitude/longitude coordinates, the other via a stereographic projection, using a diagonal RBF kernel with hyperparameters fitted with maximum likelihood) fail drastically given the high non-Gaussianity of the data. In the interpolation task, the NDP performs as well as the GEOMNDP, but the additional geometric structure of modelling the outputs living on the sphere appears to significantly help for extrapolation. See App. F.3 for more fine-grained results.

## 6  Discussion

In this work, we have extended diffusion models to model invariant stochastic processes over tensor fields. We did so by (a) constructing a continuous noising process over function spaces which correlate input samples with an equivariant kernel, (b) parameterising the score with an equivariant neural network. We have empirically demonstrated the ability of our introduced model GEOMNDP to fit complex stochastic processes, and by encoding the symmetry of the problem at hand, we show that it is more data efficient and better able to generalise.

We highlight below some current limitations and important research directions. First, evaluating the model is slow as it relies on costly SDE or ODE solvers, as existing diffusion models. Second, targeting a white noise process appears to over-perform other Gaussian processes. In future work, we would like to investigate the practical influence of different kernels. Third, strict invariance may sometimes be too strong, we thus suggest softening it by amortising the score network over extra spatial information available from the problem at hand.

Table 2: Comparative results of different models on the cyclone dataset, comparing test set likelihood, interpolation likelihood and mean squared error (MSE), and extrapolation likelihood and mean squared error. These are estimated over 5 random seeds. We only report likelihoods of models defined w.r.t the uniform measure on $\mathcal{S}^2$.

| MODEL | TEST DATA Likelihood | INTERPOLATION Likelihood | MSE (km) | EXTRAPOLATION Likelihood | MSE (km) |
|---|---|---|---|---|---|
| GEOMNDP ($\mathbb{R} \to \mathcal{S}^2$) | $\mathbf{802_{\pm 5}}$ | $\mathbf{535_{\pm 4}}$ | $\mathbf{162_{\pm 6}}$ | $\mathbf{536_{\pm 4}}$ | $\mathbf{496_{\pm 14}}$ |
| STEREOGRAPHIC GP ($\mathbb{R} \to \mathbb{R}^2/\{0\}$) | $393_{\pm 3}$ | $266_{\pm 3}$ | $2619_{\pm 13}$ | $245_{\pm 2}$ | $6587_{\pm 55}$ |
| NDP ($\mathbb{R} \to \mathbb{R}^2$) | - | - | $166_{\pm 22}$ | - | $769_{\pm 48}$ |
| GP ($\mathbb{R} \to \mathbb{R}^2$) | - | - | $6852_{\pm 41}$ | - | $8138_{\pm 87}$ |

## Acknowledgements

We are grateful to Paul Rosa for helping with the proof, and to José Miguel Hernández-Lobato for useful discussions. We thank the `hydra` (Yadan, 2019), `jax` (Bradbury et al., 2018) and `geomstats` (Miolane et al., 2020) teams, as our library is built on these great libraries. Richard E. Turner and Emile Mathieu are supported by an EPSRC Prosperity Partnership EP/T005386/1 between Microsoft Research and the University of Cambridge. Michael J. Hutchinson is supported by the EPSRC Centre for Doctoral Training in Modern Statistics and Statistical Machine Learning (EP/S023151/1).

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

# Supplementary to:
# Geometric Neural Diffusion Processes

## A  Organisation of appendices

In this supplementary, we first introduce in App. B an Ornstein Uhlenbeck process on function space (via finite marginals) along with several score approximations. Then in App. C, we show how this methodology extend to manifold-valued inputs or outputs. Later in App. D, we derive sufficient conditions for this introduced model to yield a group invariant process. What's more in App. E, we study some conditional sampling schemes. Eventually in App. F, we give a thorough description of experimental settings along with additional empirical results.

## B  Ornstein Uhlenbeck on function space

### B.1  Multivariate Ornstein-Uhlenbeck process

First, we aim to show that we can define a stochastic process on an infinite dimensional function space, by defining the joint finite marginals $\mathbf{Y}(x)$ as the solution of a multidimensional Ornstein-Uhlenbeck process. In particular, for any set of input $x = (x_1, \cdots, x_k) \in \mathcal{X}^k$, we define the joint marginal as the solution of the following SDE

$$\mathrm{d}\tilde{\mathbf{Y}}_t(x) = (m(x) - \tilde{\mathbf{Y}}_t(x))/2\,\beta_t \mathrm{d}t + \sqrt{\beta_t K(x,x)}\mathrm{d}\mathbf{B}_t \ . \tag{8}$$

**Proposition B.1.** *(Phillips et al., 2022) We assume we are given a data process $(\mathbf{Y}_0(x))_{x \in \mathcal{X}}$ and we denote by $\mathbf{G} \sim \mathrm{GP}(0, k)$ a Gaussian process with zero mean and covariance. Then let's define*

$$\mathbf{Y}_t \triangleq e^{-\frac{1}{2} \cdot \int_{s=0}^{t} \beta_s ds}\, \mathbf{Y}_0 + \left(1 - e^{-\frac{1}{2} \cdot \int_{s=0}^{t} \beta_s ds}\right) m + \left(1 - e^{-\int_{s=0}^{t} \beta_s ds}\right)^{1/2} \mathbf{G}.$$

*Then $(\mathbf{Y}_t(x))_{x \in \mathcal{X}}$ is a stochastic process (by virtue of being a linear combination of stochastic processes). We thus have that $\mathbf{Y}_t \xrightarrow[t \to 0]{a.s.} \mathbf{Y}_0$ and $\mathbf{Y}_t \xrightarrow[t \to \infty]{a.s.} \mathbf{Y}_\infty$ with $\mathbf{Y}_\infty \sim \mathrm{GP}(m, k)$, so effectively $(\mathbf{Y}_t(x))_{t \in \mathbb{R}_+, x \in \mathcal{X}}$ interpolates between the data process and this limiting Gaussian process. Additionally, $\mathcal{L}(\mathbf{Y}_t | \mathbf{Y}_0 = y_0) = \mathrm{GP}(m_t, K_t)$ with $m_t = e^{-\frac{1}{2} \cdot \int_{s=0}^{t} \beta_s ds} y_0 + \left(1 - e^{-\frac{1}{2} \cdot \int_{s=0}^{t} \beta_s ds}\right) m$ and $\Sigma_t = \left(1 - e^{-\int_{s=0}^{t} \beta_s ds}\right) K$. Furthermore, $(\mathbf{Y}_t(x))_{t \in \mathbb{R}_+, x \in \mathcal{X}}$ is the solution of the SDE in (8).*

*Proof.* We aim to compute the mean and covariance of the process $(\mathbf{Y}_t)_{t \geq 0}$ described by the SDE (3). First let's recall the time evolution of the mean and covariance of the solution from a multivariate Ornstein-Uhlenbeck process given by

$$\mathrm{d}\mathbf{Y}_t = f(\mathbf{Y}_t, t)\mathrm{d}t + L(\mathbf{Y}_t, t)\mathrm{d}\mathbf{B}_t. \tag{9}$$

We know that the time evolution of the mean and the covariance are given respectively by Särkkä and Solin (2019)

$$\frac{\mathrm{d}m_t}{\mathrm{d}t} = \mathrm{E}[f(\mathbf{Y}_t, t)] \tag{10}$$

$$\frac{\mathrm{d}\Sigma_t}{\mathrm{d}t} = \mathrm{E}[f(\mathbf{Y}_t, t)(m_t - \mathbf{Y}_t)^\top] + \mathrm{E}[(m_t - \mathbf{Y}_t)f(\mathbf{Y}_t, t)^\top] + \mathrm{E}[L(\mathbf{Y}_t, t)L(\mathbf{Y}_t, t)^\top]. \tag{11}$$

Plugging in the drift $f(\mathbf{Y}_t, t) = 1/2 \cdot (m - \mathbf{Y}_t)\beta_t$ and diffusion term $L(\mathbf{Y}_t, t) = \sqrt{\beta_t K}$ from (3), we get

$$\frac{\mathrm{d}m_t}{\mathrm{d}t} = 1/2 \cdot (m - \mathbf{Y}_t)\beta_t \tag{12}$$

$$\frac{\mathrm{d}\Sigma_t}{\mathrm{d}t} = \beta_t \left[K - \Sigma_t\right]. \tag{13}$$

Solving these two ODEs we get

$$m_t = e^{-\frac{1}{2} \cdot \int_{s=0}^t \beta_s ds} m_0 + \left(1 - e^{-\frac{1}{2} \cdot \int_{s=0}^t \beta_s ds}\right) m \tag{14}$$

$$\Sigma_t = K + e^{-\int_{s=0}^t \beta_s ds} (\Sigma_0 - K) \tag{15}$$

with $m_0 \triangleq \mathrm{E}[\mathbf{Y}_0]$ and $\Sigma_0 \triangleq \mathrm{Cov}[\mathbf{Y}_0]$.

Now let's compute the first two moments of $(\mathbf{Y}_t(x))_{x \in \mathcal{X}}$. We have

$$\mathrm{E}[\mathbf{Y}_t] = \mathrm{E}\left[e^{-\frac{1}{2} \cdot \int_{s=0}^t \beta_s ds} \mathbf{Y}_0 + \left(1 - e^{-\frac{1}{2} \cdot \int_{s=0}^t \beta_s ds}\right) m + \left(1 - e^{-\frac{1}{2} \cdot \int_{s=0}^t \beta_s ds}\right) \mathbf{G}\right] \tag{16}$$

$$= e^{-\frac{1}{2} \cdot \int_{s=0}^t \beta_s ds} m_0 + \left(1 - e^{-\frac{1}{2} \cdot \int_{s=0}^t \beta_s ds}\right) m \tag{17}$$

$$= m_t \tag{18}$$

$$\mathrm{Cov}[\mathbf{Y}_t] = \mathrm{Cov}\left[e^{-\frac{1}{2} \cdot \int_{s=0}^t \beta_s ds} \mathbf{Y}_0\right] + \mathrm{Cov}\left[\left(1 - e^{-\int_{s=0}^t \beta_s ds}\right)^{1/2} \mathbf{G}\right] \tag{19}$$

$$= e^{-\int_{s=0}^t \beta_s ds} \Sigma_0 + \left(1 - e^{-\int_{s=0}^t \beta_s ds}\right) K \tag{20}$$

$$= K + e^{-\int_{s=0}^t \beta_s ds} (\Sigma_0 - K) \tag{21}$$

$$= \Sigma_t . \tag{22}$$

$\square$

## B.2 Conditional score

Hence, condition on $\mathbf{Y}_0$ the score is the gradient of the log Gaussian characterised by mean $m_{t|0} = e^{-\frac{1}{2}B(t)}\mathbf{Y}_0$ and $\Sigma_{t|0} = (1 - e^{-B(t)})K$ with $B(t) = \int_0^t \beta(s)ds$ which can be derived from the above marginal mean and covariance with $m_0 = \mathbf{Y}_0$ and $\Sigma_0 = 0$.

$$\nabla_{\mathbf{Y}_t} \log p_t(\mathbf{Y}_t | \mathbf{Y}_0) = \nabla_{\mathbf{Y}_t} \log \mathcal{N}\left(\mathbf{Y}_t | m_{t|0}, \Sigma_{t|0}\right) \tag{23}$$

$$= \nabla_{\mathbf{Y}_t} - 1/2(\mathbf{Y}_t - m_{t|0})^\top \Sigma_{t|0}^{-1} (\mathbf{Y}_t - m_{t|0}) + c \tag{24}$$

$$= -\Sigma_{t|0}^{-1}(\mathbf{Y}_t - m_{t|0}) \tag{25}$$

$$= -\mathrm{L}_{t|0}^{-\top} \mathrm{L}_{t|0}^{-1} \mathrm{L}_{t|0} \epsilon \tag{26}$$

$$= -\mathrm{L}_{t|0}^{-\top} \epsilon \tag{27}$$

where $\mathrm{L}_{t|0}$ denotes the Cholesky decomposition of $\Sigma_{t|0} = \mathrm{L}_{t|0}\mathrm{L}_{t|0}^\top$, and $\mathbf{Y}_t = m_{t|0} + \mathrm{L}_{t|0}\epsilon$.

Then we can plugin our learnt (preconditioned) score into the backward SDE 4 which gives

$$d\bar{\mathbf{Y}}_t | x = \left[-(m(x) - \bar{\mathbf{Y}}_t)/2 + \mathrm{K}(x, x)\nabla_{\bar{\mathbf{Y}}_t} \log p_{T-t}(t, x, \bar{\mathbf{Y}}_t)\right] dt + \sqrt{\beta_t \mathrm{K}(x, x)}\beta_t d\mathbf{B}_t \tag{28}$$

## B.3 Several score parametrisations

In this section, we discuss several parametrisations of the neural network and the objective.

For the sake of versatility, we opt to employ the symbol $D_\theta$ for the network instead of $s_\theta$ as mentioned in the primary text, as it allows us to approximate not only the score but also other quantities from which the score can be derived. In full generality, we use a residual connection, weighted by $c_{\mathrm{out}}, c_{\mathrm{skip}} : \mathbb{R} \to \mathbb{R}$, to parameterise the network

$$D_\theta(t, \mathbf{Y}_t) = c_{\mathrm{skip}}(t)\mathbf{Y}_t + c_{\mathrm{out}}(t)F_\theta(t, \mathbf{Y}_t). \tag{29}$$

We recall that the input to the network is time $t$, and the noised vector $\mathbf{Y}_t = \boldsymbol{\mu}_{t|0} + \boldsymbol{n}$, where $\boldsymbol{\mu}_{t|0} = e^{-B(t)/2}\mathbf{Y}_0$ and $\boldsymbol{n} \sim \mathcal{N}(0, \Sigma_{t|0})$ with $\Sigma_{t|0} = (1 - e^{-B(t)})K$. The gram matrix $K$ corresponds to $k(X, X)$ with $k$ the limiting kernel. We denote by $\mathrm{L}_{t|0}$ and $\mathrm{S}$ respectively the Cholesky decomposition of $\Sigma_{t|0} = \mathrm{L}_{t|0}\mathrm{L}_{t|0}^\top$ and $K = \mathrm{SS}^\top$.

The denoising score matching loss weighted by $\Lambda(t)$ is given by

$$\mathcal{L}(\theta) = \mathbb{E}\left[\|D_\theta(t, \mathbf{Y}_t) - \nabla_{\mathbf{Y}_t} \log p_t(\mathbf{Y}_t | \mathbf{Y}_0)\|_{\Lambda(t)}^2\right] \tag{30}$$

Table 3: Summary of different score parametrisations as well as the values for $c_{\text{skip}}$ and $c_{\text{out}}$ that we found to be optimal, based on the recommendation from Karras et al. (2022, Appendix B.6).

| | No precond. | Precond. $K$ | Precond. $S^\top$ | Predict $\boldsymbol{Y}_0$ |
|---|---|---|---|---|
| $c_{\text{skip}}$ | 0 | 0 | 0 | 1 |
| $c_{\text{out}}$ | $(\sigma_{t\|0} + 10^{-3})^{-1}$ | $(\sigma_{t\|0} + 10^{-3})^{-1}$ | $(\sigma_{t\|0} + 10^{-3})^{-1}$ | 1 |
| Loss | $\|\sigma_{t\|0}S^\top D_\theta + \boldsymbol{z}\|_2^2$ | $\|\sigma_{t\|0}D_\theta + S\boldsymbol{z}\|_2^2$ | $\|\sigma_{t\|0}D_\theta + \boldsymbol{z}\|_2^2$ | $\|D_\theta - \boldsymbol{Y}_0\|_2^2$ |
| $K\nabla \log p_t$ | $KD_\theta$ | $D_\theta$ | $SD_\theta$ | $-\Sigma_{t\|0}^{-1}(\boldsymbol{Y}_t - e^{-\frac{B(t)}{2}}D_\theta)$ |

**No preconditioning**  By reparametrisation, let $\boldsymbol{Y}_t = \boldsymbol{\mu}_{t|0} + L_{t|0}\boldsymbol{z}$, where $\boldsymbol{z} \sim \mathcal{N}(\boldsymbol{0}, \mathbf{I})$, the loss from Eq. (30) can be written as

$$\mathcal{L}(\theta) = \mathbb{E}\left[\|D_\theta(t, \boldsymbol{Y}_t) + \Sigma_{t|0}^{-1}(\boldsymbol{Y}_t - \boldsymbol{\mu}_{t|0})\|_{\Lambda(t)}^2\right] \tag{31}$$

$$= \mathbb{E}\left[\|D_\theta(t, \boldsymbol{Y}_t) + \Sigma_{t|0}^{-1}L_{t|0}\boldsymbol{z}\|_{\Lambda(t)}^2\right] \tag{32}$$

$$= \mathbb{E}\left[\|D_\theta(t, \boldsymbol{Y}_t) + L_{t|0}^{-\top}\boldsymbol{z}\|_{\Lambda(t)}^2\right] \tag{33}$$

$$\tag{34}$$

Choosing $\Lambda(t) = \Sigma_{t|0} = L_{t|0}L_{t|0}^\top$ we obtain

$$\mathcal{L}(\theta) = \mathbb{E}\left[\|L_{t|0}^\top D_\theta(t, \boldsymbol{Y}_t) + \boldsymbol{z}\|_2^2\right] \tag{35}$$

$$= \mathbb{E}\left[\|\sigma_{t|0}S^\top D_\theta(t, \boldsymbol{Y}_t) + \boldsymbol{z}\|_2^2\right]. \tag{36}$$

**Preconditioning by $K$**  Alternatively, one can train the neural network to approximate the preconditioned score $D_\theta \approx \mathbf{K}\nabla_{\boldsymbol{Y}_t}\log p_t(\boldsymbol{Y}_t|\boldsymbol{Y}_0)$. The loss, weighted by $\Lambda = \sigma_{t|0}^2\mathbf{I}$, is then given by

$$\mathcal{L}(\theta) = \mathbb{E}\left[\|D_\theta(t, \boldsymbol{Y}_t) + K L_{t|0}^{-\top}\boldsymbol{z}\|_{\Lambda(t)}^2\right] \tag{37}$$

$$= \mathbb{E}\left[\|D_\theta(t, \boldsymbol{Y}_t) + \sigma_{t|0}^{-1}S\boldsymbol{z}\|_{\Lambda(t)}^2\right] \tag{38}$$

$$= \mathbb{E}\left[\|\sigma_{t|0}D_\theta(t, \boldsymbol{Y}_t) + S\boldsymbol{z}\|_2^2\right]. \tag{39}$$

**Precondition by $S^\top$**  A variation of the previous one, is to precondition the score by the transpose Cholesky of the limiting kernel gram matrix, such that $D_\theta \approx S^\top \nabla_{\boldsymbol{Y}_t}\log p_t(\boldsymbol{Y}_t|\boldsymbol{Y}_0)$.

The loss, weighted by $\Lambda = \sigma_{t|0}^2\mathbf{I}$, becomes

$$\mathcal{L}(\theta) = \mathbb{E}\left[\|D_\theta(t, \boldsymbol{Y}_t) + S^\top L_{t|0}^{-\top}\boldsymbol{z}\|_{\Lambda(t)}^2\right] \tag{40}$$

$$= \mathbb{E}\left[\|D_\theta(t, \boldsymbol{Y}_t) + \sigma_{t|0}^{-1}\boldsymbol{z}\|_{\Lambda(t)}^2\right] \tag{41}$$

$$= \mathbb{E}\left[\|\sigma_{t|0}D_\theta(t, \boldsymbol{Y}_t) + \boldsymbol{z}\|_2^2\right]. \tag{42}$$

**Predicting $\boldsymbol{Y}_0$**  Finally, an alternative strategy is to predict $\boldsymbol{Y}_0$ from a noised version $\boldsymbol{Y}_t$. In this case, the loss takes the simple form

$$\mathcal{L}(\theta) = \mathbb{E}\left[\|D_\theta(t, \boldsymbol{Y}_t) - \boldsymbol{Y}_0\|_2^2\right].$$

The score can be computed from the network's prediction following

$$\nabla \log p_t(\boldsymbol{Y}_t|\boldsymbol{Y}_0) = -\Sigma_{t|0}^{-1}(\boldsymbol{Y}_t - \boldsymbol{\mu}_{t|0}) \tag{43}$$

$$= -\Sigma_{t|0}^{-1}(\boldsymbol{Y}_t - e^{-B(t)/2}\boldsymbol{Y}_0) \tag{44}$$

$$\approx -\Sigma_{t|0}^{-1}\left(\boldsymbol{Y}_t - e^{-B(t)/2}D_\theta(t, \boldsymbol{Y}_t)\right) \tag{45}$$

$$\tag{46}$$

Table 3 summarises the different options for parametrising the score as well as the values for $c_{\text{skip}}$ and $c_{\text{out}}$ that we found to be optimal, based on the recommendation from Karras et al. (2022, Appendix B.6). In practice, we found the precondition by $K$ parametrisation to produce the best results, but we refer to App. F.1.3 for a more in-depth ablation study.

## B.4 Exact (marginal) score in Gaussian setting

Interpolating between Gaussian processes $GP(m_0, \Sigma_0)$ and $GP(m, \mathrm{K})$

$$\mathrm{K}\nabla_{\bar{\mathbf{Y}}_t} \log p_t(\mathbf{Y}_t) = -\mathrm{K}\Sigma_t^{-1}(\mathbf{Y}_t - m_t) \tag{47}$$

$$= -\mathrm{K}[\mathrm{K} + e^{-\int_{s=0}^{t} \beta_s ds}(\Sigma_0 - \mathrm{K})]^{-1}(\mathbf{Y}_t - m_t) \tag{48}$$

$$= -\mathrm{K}(\mathrm{L}_t \mathrm{L}_t^\top)^{-1}(\mathbf{Y}_t - m_t) \tag{49}$$

$$= -\mathrm{K}\mathrm{L}_t^{\top-1}\mathrm{L}_t^{-1}(\mathbf{Y}_t - m_t) \tag{50}$$

$$\tag{51}$$

with $\Sigma_t = \mathrm{K} + e^{-\int_{s=0}^{t} \beta_s ds}(\Sigma_0 - \mathrm{K}) = \mathrm{L}_t \mathrm{L}_t^\top$ obtained via Cholesky decomposition.

## B.5 Langevin dynamics

Under mild assumptions on $\nabla \log p_{T-t}$ (Durmus and Moulines, 2016) the following SDE

$$\mathrm{d}\mathbf{Y}_s = \tfrac{1}{2}\mathrm{K}\nabla \log p_{T-t}(\mathbf{Y}_s)\mathrm{d}s + \sqrt{\mathrm{K}}\mathrm{d}\mathbf{B}_s \tag{52}$$

admits a solution $(\mathbf{Y}_s)_{s \geq 0}$ whose law $\mathcal{L}(\mathbf{Y}_s)$ converges with geometric rate to $p_{T-t}$ for any invertible matrix K.

## B.6 Likelihood evaluation

Similarly to Song et al. (2021), we can derive a deterministic process which has the same marginal density as the SDE (3), which is given by the following Ordinary Differential Equation (ODE)— referred as the probability flow ODE

$$\mathrm{d}\begin{pmatrix} \mathbf{Y}_t(x) \\ \log p_t(\mathbf{Y}_t(x)) \end{pmatrix} = \begin{pmatrix} \tfrac{1}{2}\{m(x) - \mathbf{Y}_t(x) - \mathrm{K}(x,x)\nabla \log p_t(\mathbf{Y}_t(x))\}\beta_t \\ -\tfrac{1}{2}\mathrm{div}\{m(x) - \mathbf{Y}_t(x) - \mathrm{K}(x,x)\nabla \log p_t(\mathbf{Y}_t(x))\}\beta_t \end{pmatrix}\mathrm{d}t. \tag{53}$$

Once the score network is learnt, we can thus use it in conjunction with an ODE solver to compute the likelihood of the model.

## B.7 Discussion consistency

So far we have defined a generative model over functions via its finite marginals $\bar{\mathbf{Y}}_T^\theta(x)$. These finite marginals were to arise from a stochastic process if, as per the Kolmogorov extension theorem (Øksendal, 2003), they satisfy *exchangeability* and *consistency* conditions. Exchangeability can be satisfied by parametrising the score network such that the score network is equivariant w.r.t permutation, i.e. $\mathbf{s}_\theta(t, \sigma \circ x, \sigma \circ y) = \sigma \circ \mathbf{s}_\theta(t, x, y)$ for any $\sigma \in \Sigma_n$. Additionally, we have that the noising process $(\mathbf{Y}_t(x))_{x \in \mathcal{X}}$ is trivially consistent for any $t \in \mathbb{R}_+$ since it is a stochastic process as per Prop. B.1, and consequently so is the (true) time-reversal $(\bar{\mathbf{Y}}_t(x))_{x \in \mathcal{X}}$. Yet, when approximating the score $\mathbf{s}_\theta \approx \nabla \log p_t$, we lose the consistency over the generative process $\bar{\mathbf{Y}}_t^\theta(x)$ as the constraint on the score network is non trivial to satisfy. This is actually a really strong constraint on the model class, and as soon as one goes beyond linearity (of the posterior w.r.t. the context set), it is non trivial to enforce without directly parameterising a stochastic process, e.g. as Phillips et al. (2022). There thus seems to be a strong trade-off between satisfying consistency, and the model's ability to fit complex process and scale to large datasets.

# C Manifold-valued diffusion process

## C.1 Manifold-valued inputs

In the main text we dealt with a simplified case of tensor fields where the tensor fields are over Euclidean space. Nevertheless, it is certainly possible to apply our methods to these settings. Significant work has been done on performing convolutions on feature fields on generic manifolds (a superset of tensor fields on generic manifolds), core references being (Cohen, 2021) for the case of homogeneous spaces and (Weiler et al., 2021) for more general Riemannian manifolds. We

recommend these as excellent mathematical introductions to the topic and build on them to describe how to formulate diffusion models over these spaces.

**Tensor fields as sections of bundles.** Formally the fields we are interested in modelling are sections $\sigma$ of associated tensor bundles of the principle $G$-bundle on a manifold $M$. We shall denote such a bundle $BM$ and the space of sections $\Gamma(BM)$. The goal, therefore, is to model *random elements* from this space of sections. For a clear understanding of this definition, please see Weiler et al. (2021, pages 73-95) for an introduction suitable to ML audiences. Prior work looking at this setting is (Hutchinson et al., 2021) where they construct Gaussian Processes over tensor fields on manifolds.

**Stochastic processes on spaces of sections.** Given we can see sections as maps $\sigma : M \to BM$, where an element in $BM$ is a tuple $(m, b)$, $m$ in the base manifold and $b$ in the typical fibre, alongside the condition that the composition of the projection $\mathrm{proj}_i : (m, b) \mapsto m$ with the section is the identity, $\mathrm{proj}_i \circ \sigma = \mathrm{Id}$ it is clear we can see distribution over sections as stochastic processes with index set the manifold $M$, and output space a point in the bundle $BM$, with the projection condition satisfied. The projection onto finite marginals, i.e. a finite set of points in the manifold, is defined as $\pi_{m_1,...,m_n}(\sigma) = (\sigma(m_1), ..., \sigma(m_n))$.

**Noising process.** To define a noising process over these marginals, we can use Gaussian Processes defined in (Hutchinson et al., 2021) over the tensor fields. The convergence results of Phillips et al. (2022) hold still, and so using these Gaussian Processes as noising processes on the marginals also defines a noising process on the whole section.

**Reverse process.** The results of (Cattiaux et al., 2021) are extremely general and continue to hold in this case of SDEs on the space of sections. Note we don't actually need this to be the case, we can just work with the reverse process on the marginals themselves, which are much simpler objects. It is good to know that it is a valid process on full sections though should one want to try and parameterise a score function on the whole section akin to some other infinite-dimension diffusion models.

**Score function.** The last thing to do therefore is parameterise the score function on the marginals. If we were trying to parameterise the score function over the *whole* section at once (akin to a number of other works on infinite dimension diffusions), this could present some problems in enforcing the smoothness of the score function. As we only deal with the score function on a finite set of marginals, however, we need not deal with this issue and this presents a distinct advantage in simplicity for our approach. All we need to do is pick a way of numerically representing points on the manifold and b) pick a basis for the tangent space of each point on the manifold. This lets us represent elements from the tangent space numerically, and therefore also elements from tensor space at each point numerically as well. This done, we can feed these to a neural network to learn to output a numerical representation of the score on the same basis at each point.

### C.2   Manifold-valued outputs

In the setting, where one aim to model a stochastic process with manifold codomain $\mathbf{Y}_t(x) = (\mathbf{Y}_t(x_1), \cdots, \mathbf{Y}_t(x_n)) \in \mathcal{M}^n$, things are less trivial as manifolds do not have a vector space structure which is necessary to define Gaussian processes. Fortunately, We can still target a know distribution marginally independently on each marginal, since this is well defined, and as such revert to the Riemannian diffusion models introduced in De Bortoli et al. (2021) with $n$ independent Langevin noising processes

$$\mathrm{d}\mathbf{Y}_t(x_k) = -\tfrac{1}{2}\nabla U(\mathbf{Y}_t(x_k))\,\beta_t \mathrm{d}t + \sqrt{\beta_t}\mathrm{d}\mathbf{B}_t^{\mathcal{M}} \ . \tag{54}$$

are applied to each marginal. Hence in the limit $t \to \infty$, $\mathbf{Y}_t(x)$ has density (assuming it exists) which factors as $dp/d\mathrm{Vol}_{\mathcal{M}}((y(x_1), \cdots, y(x_n))) \propto \prod_k e^{-U(y(x_n)))}$. For compact manifolds, we can target the uniform distribution by setting $U(x) = 0$. The reverse time process will have correlation between different marginals, and so the score function still needs to be a function of all the points in the marginal of interest.

## D   Invariant neural diffusion processes

### D.1   E$(n)$-equivariant kernels

A kernel $k : \mathbb{R}^d \times \mathbb{R}^d \to \mathbb{R}^{d \times d}$ is equivariant if it satisfies the following constraints: (a) $k$ is *stationnary*, that is if for all $x, x' \in \mathbb{R}^n$

$$k(x, x') = k(x - x') \triangleq \tilde{k}(x - x') \tag{55}$$

and if (b) it satisfies the *angular constraint* for any $h \in H$

$$k(hx, hx') = \rho(h)k(x, x')\rho(h)^\top. \tag{56}$$

A trivial example of such an equivariant kernel is the diagonal kernel $k(x, x') = k_0(x, x')\mathrm{I}$ (Holderrieth et al., 2021), with $k_0$ stationnary. This kernel can be understood has having $d$ independent Gaussian process uni-dimensional output, that is, there is no inter-dimensional correlation.

Less trivial examples, are the $\mathrm{E}(n)$ equivariant kernels proposed in Macêdo and Castro (2010). Namely curl-free and divergence-free kernels, allowing for instance to model electric or magnetic fields. Formally we have $k_{\mathrm{curl}} = k_0 A$ and $k_{\mathrm{div}} = k_0 B$ with $k_0$ stationary, e.g. squared exponential kernel $k_0(x, x') = \sigma^2 \exp\left(\frac{\|x - x'\|^2}{2l^2}\right)$, and $A$ and $B$ given by

$$A(x, x') = \mathrm{I} - \frac{(x - x')(x - x')^\top}{l^2} \tag{57}$$

$$B(x, x') = \frac{(x - x')(x - x')^\top}{l^2} + \left(n - 1 - \frac{\|x - x'\|^2}{l^2}\right)\mathrm{I}. \tag{58}$$

See Holderrieth et al. (Appendix C, 2021) for a proof.

### D.2 Proof of Prop. 3.2

Below we give two proofs for the group invariance of the generative process, one via the probability flow ODE and one directly via Fokker-Planck.

*Proof.* Reverse ODE. The reverse probability flow associated with the forward SDE (3) with approximate score $\mathbf{s}_\theta(t, \cdot) \approx \nabla \log p_t$ is given by

$$d\bar{\mathbf{Y}}_t | x = \tfrac{1}{2}\left[-m(x) + \bar{\mathbf{Y}}_t + \mathrm{K}(x, x)\mathbf{s}_\theta(T - t, x, \bar{\mathbf{Y}}_t)\right]dt \tag{59}$$

$$\triangleq b_{\mathrm{ODE}}(t, x, \bar{\mathbf{Y}}_t)dt \tag{60}$$

This ODE induces a flow $\phi_t^b : X^n \times Y^n \to TY^n$ for a given integration time $t$, which is said to be $G$-equivariant if the vector field is $G$-equivariant itself, i.e. $b(t, g \cdot x, \rho(g)\bar{\mathbf{Y}}_t) = \rho(g)b(t, x, \bar{\mathbf{Y}}_t)$. We have that for any $g \in G$

$$b_{\mathrm{ODE}}(t, g \cdot x, \rho(g)\bar{\mathbf{Y}}_t) = \tfrac{1}{2}\left[-m(g \cdot x) + \rho(g)\mathbf{Y}_t + \mathrm{K}(g \cdot x, g \cdot x)\,\mathbf{s}_\theta(t, g \cdot x, \rho(g)\bar{\mathbf{Y}}_t)\right] \tag{61}$$

$$\overset{(1)}{=} \tfrac{1}{2}\left[-\rho(g)m(x) + \rho(g)\mathbf{Y}_t + \rho(g)\mathrm{K}(x, x)\rho(g)^\top\,\mathbf{s}_\theta(t, g \cdot x, \rho(g)\bar{\mathbf{Y}}_t)\right] \tag{62}$$

$$\overset{(2)}{=} \tfrac{1}{2}\left[-\rho(g)m(x) + \rho(g)\mathbf{Y}_t + \rho(g)\mathrm{K}(x, x)\rho(g)^\top\rho(g)\,\mathbf{s}_\theta(t, x, \bar{\mathbf{Y}}_t)\right] \tag{63}$$

$$\overset{(3)}{=} \tfrac{1}{2}\rho(g)\left[-m(x) + \mathbf{Y}_t + \mathrm{K}(x, x)\,\mathbf{s}_\theta(t, x, \bar{\mathbf{Y}}_t)\right] \tag{64}$$

$$= \rho(g)b_{\mathrm{ODE}}(t, x, \bar{\mathbf{Y}}_t) \tag{65}$$

with (1) from the $G$-invariant prior GP conditions on $m$ and $k$, (2) assuming that the score network is $G$-equivariant and (3) assuming that $\rho(g) \in O(n)$. To prove the opposite direction, we can simply follow these computations backwards. Finally, we know that with a $G$-invariant probability measure $p_{\mathrm{ref}}$ and $G$-equivariant map $\phi$, the pushforward probability measure $p_{\mathrm{ref}}^{-1} \circ \phi$ is also $G$-invariant (Köhler et al., 2020; Papamakarios et al., 2019). Assuming a $G$-invariant prior GP, and a $G$-equivariant score network, we thus have that the generative model from Sec. 3 defines marginals that are $G$-invariant. $\square$

*Proof.* Reverse SDE. The reverse SDE associated of the forward SDE (3) with approximate score $\mathbf{s}_\theta(t, \cdot) \approx \nabla \log p_t$ is given by

$$d\bar{\mathbf{Y}}_t | x = \left[-(m(x) - \bar{\mathbf{Y}}_t)/2 + \mathrm{K}(x, x)\mathbf{s}_\theta(T - t, x, \bar{\mathbf{Y}}_t)\right]dt + \sqrt{\beta_t \mathrm{K}(x, x)}d\mathbf{B}_t \tag{66}$$

$$\triangleq b_{\mathrm{SDE}}(t, x, \bar{\mathbf{Y}}_t)dt + \Sigma^{1/2}(t, x)\,d\mathbf{B}_t. \tag{67}$$

As for the probability flow drift $b_{\text{ODE}}$, we have that $b_{\text{SDE}}$ is similarly $G$-equivariant, that is $b_{\text{SDE}}(t, g \cdot x, \rho(g)\bar{\mathbf{Y}}_t) = \rho(g)b_{\text{SDE}}(t, x, \bar{\mathbf{Y}}_t)$ for any $g \in G$. Additionally, we have that diffusion matrix is also $G$-equivariant as for any $g \in G$ we have $\Sigma(t, g \cdot x) = \beta_t K(g \cdot x, g \cdot x) = \beta_t \rho(g)K(x, x)\rho(g)^\top = \rho(g)\Sigma(t, x)\rho(g)^\top$ since K is the gram matrix of an $G$-equivariant kernel $k$.

Additionally assuming that $b_{\text{SDE}}$ and $\Sigma$ are bounded, Yim et al. (Proposition 3.6, 2023) says that the distribution of $\bar{\mathbf{Y}}_t$ is $G$-invariant, and in in particular $\mathcal{L}(\bar{\mathbf{Y}}_0)$.

□

### D.3 Equivariant posterior maps

**Theorem D.1** (Invariant prior stochastic process implies an equivariant posterior map). *Using the language of Weiler et al. (2021) our tensor fields are sections of an associated vector bundle $\mathcal{A}$ of a manifold $M$ with a $G$ structure. Let $\text{Isom}_{GM}$ be the group of $G$-structure preserving isometries on $M$. The action of this group on a section of the bundle $f \in \Gamma(\mathcal{A})$ is given by*

$$\phi \rhd f := \phi_{*,\mathcal{A}} \circ f \circ \phi^{-1}$$

*(Weiler et al., 2021). Let $f \sim P$, $P$ a distribution over the space of section. Let $\phi \rhd P$ be the law of of $\phi \rhd f$. Let $\mu_x = \mathcal{L}(f(x)) = \pi_{x\#}P$, the law of $f$ evaluated at a point, where $\pi_x$ is the canonical projection operator onto the marginal at $x$, $\#$ the pushforward operator in the measure theory sense, $x \in M$ and $y$ is in the fibre of the associated bundle. Let $\mu_x^{x',y} = \mathcal{L}(f(x)|f(x') = y') = \pi_x \mu^{x',y'} = \pi_{x\#}\mathcal{L}(f|f(x') = y')$, the conditional law of the process when given $f(x') = y'$.*

*Assume that the prior is invariant under the action of $\text{Isom}_{GM}$, i.e. that*

$$\phi \rhd \mu_x = (\phi_{*,\mathcal{A}})_\# \mu_{\phi^{-1}(x)} = \mu_x$$

*Then the conditional measures are equivariant, in the sense that*

$$\phi \rhd \mu_x^{x',y'} = (\phi_{*,\mathcal{A}})_\# \mu_{\phi^{-1}(x)}^{x',y'} = \mu_x^{\phi^{-1}(x),\phi_{*,\mathcal{A}}(y)} = \mu_x^{\phi\rhd(x',y')}$$

*Proof.* $\forall A, B$ test functions, $\phi \in \text{Isom}_{GM}$,

$$
\begin{aligned}
\mathrm{E}[B(f(x'))A((\phi \rhd f)(x))] &= \mathrm{E}\big[B(f(x'))A\big(\phi_{*,\mathcal{A}} \circ f \circ \phi^{-1}(x)\big)\big] \\
&= \mathrm{E}\big[B(f(x'))\mathrm{E}\big[A\big(\phi_{*,\mathcal{A}}\big(F(\phi^{-1}(x))\big)\big) \mid F(x')\big]\big] \\
&= \mathrm{E}\bigg[B(f(x'))\int A(y)(\phi_{*,\mathcal{A}})_\# \mu_{\phi^{-1}(x)}^{x',f(x')}(\mathrm{d}y)\bigg] \\
&= \int B(y')\int A(y)(\phi_{*,\mathcal{A}})_\# \mu_{\phi^{-1}(x)}^{x',f(x')}(\mathrm{d}y)\mu_{x'}(\mathrm{d}y') \\
&= \int B(y')\int A(y)\big(\phi \rhd \mu_x^{x',f(x')}\big)(\mathrm{d}y)\mu_{x'}(\mathrm{d}y')
\end{aligned}
$$

By invariance this quantity is also equal to

$$
\begin{aligned}
\mathrm{E}\big[B\big((\phi^{-1} \rhd f)(x')\big)A\big((\phi^{-1} \rhd \phi \rhd f)(x)\big)\big] &= \mathrm{E}\big[B\big((\phi^{-1} \rhd f)(x')\big)\mathrm{E}\big[A(f(x)) \mid B\big((\phi^{-1} \rhd f)(x')\big)\big]\big] \\
&= \mathrm{E}\big[B\big(\phi_{*,\mathcal{A}}(f(\phi^{-1}(x')))\big)\big[A(F(x)) \mid \phi_{*,\mathcal{A}}(f(\phi^{-1}(x')))\big]\big] \\
&= \mathrm{E}\bigg[B\Big(\tau_{x',g}^{-1}F(gx')\Big)\int A(y)\mu_x^{\phi(x'),\phi_{*,\mathcal{A}}^{-1}(y)}\bigg](\mathrm{d}y) \\
&= \int B(y')\int A(y)\mu_x^{\phi\rhd(x',y)}(\mathrm{d}y)\Big(\phi_{*,\mathcal{A}}^{-1}\Big)_\# \mu_{\phi(x')}(\mathrm{d}y') \\
&= \int B(y')\int A(y)\mu_x^{\phi\rhd(x',y)}(\mathrm{d}y)\big(\phi^{-1} \rhd \mu_{x'}\big)(\mathrm{d}y')
\end{aligned}
$$

Hence

$$\big(\phi \rhd \mu_x^{x',f(x')}\big)(\mathrm{d}y)\mu_{x'}(\mathrm{d}y') = \mu_x^{\phi\rhd(x',y)}(\mathrm{d}y)\big(\phi^{-1} \rhd \mu_{x'}\big)(\mathrm{d}y')$$

By the stated invariance $\phi^{-1} \triangleright \mu_{x'} = \mu_{x'}$, hence

$$\left( \phi \triangleright \mu_x^{x',f(x')} \right)(\mathrm{d}y) = \mu_x^{\phi \triangleright (x',y)}(\mathrm{d}y) \text{ a.e. } y'$$

So

$$\phi \triangleright \mu_x^{x',f(x')} = \mu_x^{\phi \triangleright (x',y)} \tag{68}$$

as desired. $\qquad\square$

## E  Langevin corrector and the iterative procedure of REPAINT (Lugmayr et al., 2022)

### E.1  Langevin sampling scheme

Several previous schemes exist for conditional sampling from Diffusion models. Two different types of conditional sampling exist. Those that try to sample conditional on some part of the state space over which the diffusion model has been trained, such as in-painting or extrapolation tasks, and those that post-hoc attempt to condition on something outside the state space that the model has been trained on.

This first category is the one we are interested in, and in it we have:

- Replacement sampling (Song et al., 2021), where the reverse ODE or SDE is evolved but by fixing the conditioning data during the rollout. This method does produce visually coherent sampling in some cases, but is not an exact conditional sampling method.
- SMC-based methods (Trippe et al., 2022), which are an exact method up to the particle filter assumption. These can produce good results but can suffer from the usual SMC methods downsides on highly multi-model data such as particle diversity collapse.
- The RePaint scheme of (Lugmayr et al., 2022). While not originally proposed as an exact sampling scheme, we will show later that it can in fact be shown that this method is doing a specific instantiation of our newly proposed method, and is therefore exact.
- Amortisation methods, e.g. Phillips et al. (2022). While they can be effective, these methods can never perform exact conditional sampling, by definition.

Our goal is to produce an exact sampling scheme that does not rely on SMC-based methods. Instead, we base our method on Langevin dynamics. If we have a score function trained over the state space $\boldsymbol{x} = [\boldsymbol{x}^c, \boldsymbol{x}^*]$, where $\boldsymbol{x}^c$ are the points we wish to condition on and $\boldsymbol{x}_s$ points we wish to sample, we exploit the following score breakdown:

$$\nabla_{\boldsymbol{x}^*} \log p(\boldsymbol{x}^*|\boldsymbol{x}^c) = \nabla_{\boldsymbol{x}^*} \log p([\boldsymbol{x}^*, \boldsymbol{x}^c]) - \nabla_{\boldsymbol{x}^*} \log p(\boldsymbol{x}^c) = \nabla_{\boldsymbol{x}^*} \log p(\boldsymbol{x})$$

If we have access to the score on the joint variables, we, therefore, have access to the conditional score by simply only taking the gradient of the joint score for the variable we are not conditioning on.

Given we have learnt $s_\theta(t, \boldsymbol{x}) \approx \nabla_{\boldsymbol{x}} \log p_t(\boldsymbol{x})$, we could use this to perform Langevin dynamics at $t = \epsilon$, some time very close to 0. Similar to (Song and Ermon, 2019) however, this produces the twin issues of how to initialise the dynamics, given a random initialisation will start the sampler in a place where the score has been badly learnt, producing slow and inaccurate sampling.

Instead, we follow a scheme of tempered Langevin sampling detailed in Alg. 1. Starting at $t = T$ we sample an initialisation of $\boldsymbol{y}^*$ based on the reference distribution. Progressing from $t = T$ towards $t = \epsilon$ we alternate between running a series of Lavgevin corrector steps to sample from the distribution $p_{t,\boldsymbol{x}^*}(\boldsymbol{y}^*|\boldsymbol{y}^c)$, and a single backwards SDE step to sample from $p_{\boldsymbol{x}}(\boldsymbol{y}_{t-\gamma}|\boldsymbol{y}_t)$ with a step size $\gamma$. At each inner and outer step, we sample a noised version of the conditioning points $\boldsymbol{y}^c$ based forward SDE applying noise to these context points, $p_{t,\boldsymbol{x}^c}(\boldsymbol{y}_t^c|\boldsymbol{y}^c)$. For the exactness of this scheme, all that matters is that at the end of the sampling scheme, we are sampling from $p_{\boldsymbol{x}^*}(\boldsymbol{y}^*|\boldsymbol{y}^c)$ (up to the $\epsilon$ away from zero clipping of the SDE). The rest of the scheme is designed to map from the initial sample at $t = T$ of $\boldsymbol{y}^*$ to a viable sample through *regions where the score has been learnt well*.

Given the noising scheme applied to the context points does not actually play into the theoretical exactness of the scheme, only the practical difficulty of staying near regions of well-learnt score, we could make a series of different choices for how to noise the context set at each step.

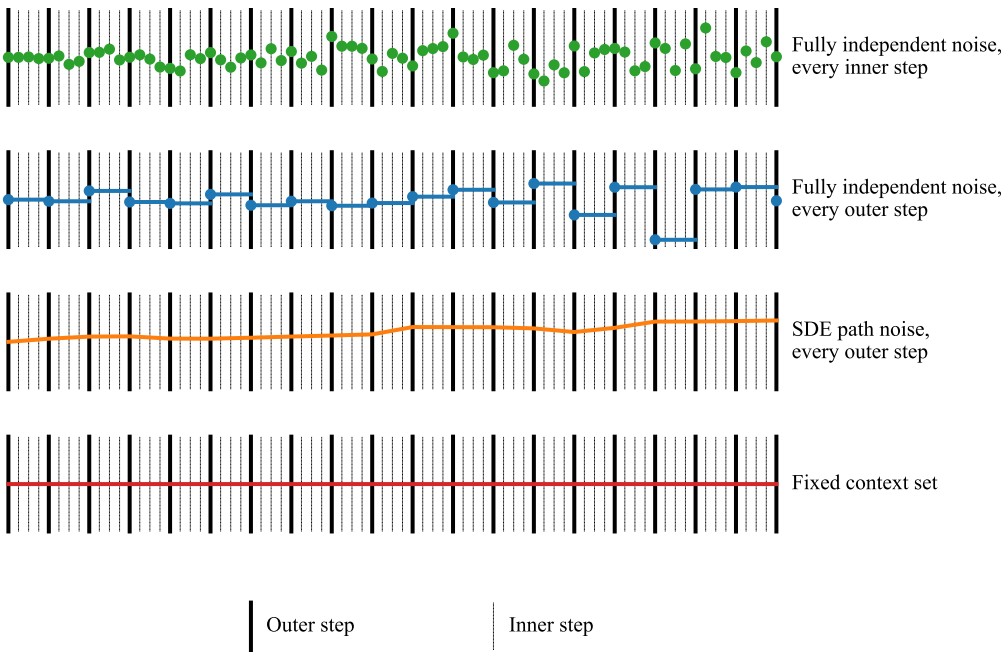

Figure 8: Comparison of different context noising schemes for the conditional sampling.

Table 4: Comparison of complexity of different noise sampling schemes for the context set.

| Scheme | Closed-form noise | Simulated noise |
|---|---|---|
| Re-sample noise at every inner step | $\mathcal{O}(NI)$ | $\mathcal{O}(N^2 I^2)$ |
| Re-sample noise at every outer step | $\mathcal{O}(N)$ | $\mathcal{O}(N^2)$ |
| Sampling an SDE path on the context | $\mathcal{O}(N)$ | $\mathcal{O}(N)$ |
| No noise applied | - | - |

The choices that present themselves are

1. The initial scheme of sampling context noise from the SDE every inner and outer step.

2. Only re-sampling the context noise every outer step, and keeping it fixed to this for each inner step associated with the outer step.

3. Instead of sampling independent marginal noise at each outer step, sampling a single noising trajectory of the context set from the forward SDE and use this as the noise at each time.

4. Perform no noising at all. Effectively the replacement method with added Langevin sampling.

These are illustrated in Fig. 8. The main trade-off of different schemes is the speed at which the noise can be sampled vs sample diversity. In the Euclidean case, we have a closed form for the evolution of the marginal density of the context point under the forward SDE. In this case sampling the noise at a given time is $\mathcal{O}(1)$ cost. On the other hand, in some instances such as nosing SDEs on general manifolds, we have to simulate this noise by discretising the forward SDE. In this case, it is $\mathcal{O}(n)$ cost, where $n$ is the number of discretisation steps in the SDE. For $N$ outer steps and $I$ inner steps, the complexity of the different noising schemes is compared in Table 4. Note the conditional sampling scheme other than the noise sampling is $\mathcal{O}(NI)$ complexity.

## E.2   REPAINT (Lugmayr et al., 2022) correspondance

In this section, we show that:

**Algorithm 1** Conditional sampling with Langevin dynamics.

---

**Require:** Score network $\mathbf{s}_\theta(t, \boldsymbol{x}, \boldsymbol{y})$, conditioning points $(\boldsymbol{x}^c, \boldsymbol{y}^c)$, query locations $\boldsymbol{x}^*$

$\bar{\boldsymbol{x}} = [\boldsymbol{x}^c, \boldsymbol{x}^*]$      ▷ Augmented inputs set

$\tilde{\boldsymbol{y}}_T^* \sim \mathrm{N}(m(\boldsymbol{x}^*), k(\boldsymbol{x}^*, \boldsymbol{x}^*))$      ▷ Sample initial noise

**for** $t \in \{T, T - \gamma, ..., \epsilon\}$ **do**

    $\boldsymbol{y}_t^c \sim p_{t, \boldsymbol{x}^c}(\boldsymbol{y}_t^c | \boldsymbol{y}_0^c)$      ▷ Noise context outputs

    $Z \sim \mathcal{N}(0, \mathrm{Id})$      ▷ Sample tangent noise

    $\left[\_, \tilde{\boldsymbol{y}}_{t-\gamma}^*\right] = [\boldsymbol{y}_t^c, \boldsymbol{y}_t^*] + \gamma \left\{ -\frac{1}{2} \left( m(\bar{\boldsymbol{x}}) - [\boldsymbol{y}_t^c, \boldsymbol{y}_t^*] \right) + \mathrm{K}(\bar{\boldsymbol{x}}, \bar{\boldsymbol{x}}) \mathbf{s}_\theta(t, \bar{\boldsymbol{x}}, [\boldsymbol{y}_t^c, \tilde{\boldsymbol{y}}_t^*]) \right\} + \sqrt{\gamma} \mathrm{K}(\bar{\boldsymbol{x}}, \bar{\boldsymbol{x}})^{1/2} Z$ ▷ Euler-Maruyama step

    **for** $l \in \{1, \ldots, L\}$ **do**

        $\boldsymbol{y}_{t-\gamma}^c \sim p_{t-\gamma, \boldsymbol{x}^c}(\boldsymbol{y}_{t-\gamma}^c | \boldsymbol{y}_0^c)$      ▷ Noise context outputs

        $Z \sim \mathcal{N}(0, \mathrm{Id})$      ▷ Sample tangent noise

        $\left[\_, \tilde{\boldsymbol{y}}_{t-\gamma}^*\right] = \left[\_, \tilde{\boldsymbol{y}}_{t-\gamma}^*\right] + \frac{\gamma}{2} \mathrm{K}(\bar{\boldsymbol{x}}, \bar{\boldsymbol{x}}) \mathbf{s}_\theta(t - \gamma, \bar{\boldsymbol{x}}, [\boldsymbol{y}_{t-\gamma}^c, \tilde{\boldsymbol{y}}_{t-\gamma}^*]) + \sqrt{\gamma} \mathrm{K}(\bar{\boldsymbol{x}}, \bar{\boldsymbol{x}})^{1/2} Z$ ▷ Langevin step

    $\boldsymbol{y}_{t-\gamma}^* = \tilde{\boldsymbol{y}}_{t-\gamma}^*$

**return** $\boldsymbol{y}_\epsilon^*$

---

(a) Alg. 1 and Alg. 2 `Repaint` from (Lugmayr et al., 2022) are equivalent in a specific setting.

(b) There exists a continuous limit (SDE) for both procedures. This SDE targets a probability density which *does not* correspond to $p(x_{t_0} | x_0^c)$.

(c) When $t_0 \to 0$ this probability measure converges to $p(x_0 | x_0^c)$ which ensures the correctness of the proposed sampling scheme.

We begin by recalling the conditional sampling algorithm we study in Alg. 1 and Alg. 2.

---

**Algorithm 2** REPAINT (Lugmayr et al., 2022).

---

**Require:** Score network $\mathbf{s}_\theta(t, \boldsymbol{x}, \boldsymbol{y})$, conditioning points $(\boldsymbol{x}^c, \boldsymbol{y}^c)$, query locations $\boldsymbol{x}^*$

$\bar{\boldsymbol{x}} = [\boldsymbol{x}^c, \boldsymbol{x}^*]$      ▷ Augmented inputs set

$[\boldsymbol{y}_T^c, \boldsymbol{y}_T^*] \sim \mathrm{N}(m(\bar{\boldsymbol{x}}), k(\bar{\boldsymbol{x}}, \bar{\boldsymbol{x}}))$      ▷ Sample initial noise

**for** $t \in \{T, T - \gamma, ..., \epsilon\}$ **do**

    $\tilde{\boldsymbol{y}}_t^* = \boldsymbol{y}_t^*$

    **for** $l \in \{1, \ldots, L\}$ **do**

        $\boldsymbol{y}_t^c \sim \mathrm{N}(m_t(\boldsymbol{x}^c; \boldsymbol{y}^c), k_t(\boldsymbol{x}^c, \boldsymbol{x}^c; \boldsymbol{y}^c))$      ▷ Noise context outputs

        $Z \sim \mathrm{N}(0, \mathrm{Id})$      ▷ Sample tangent noise

        $\left[\_, \tilde{\boldsymbol{y}}_{t-\gamma}^*\right] = [\boldsymbol{y}_t^c, \tilde{\boldsymbol{y}}_t^*] + \gamma \left\{ -\frac{1}{2} \left( m(\bar{\boldsymbol{x}}) - [\boldsymbol{y}_t^c, \tilde{\boldsymbol{y}}_t^*] \right) + \mathrm{K}(\bar{\boldsymbol{x}}, \bar{\boldsymbol{x}}) \mathbf{s}_\theta(t, \bar{\boldsymbol{x}}, [\boldsymbol{y}_t^c, \tilde{\boldsymbol{y}}_t^*]) \right\} + \sqrt{\gamma} \mathrm{K}(\bar{\boldsymbol{x}}, \bar{\boldsymbol{x}})^{1/2} Z$ ▷ Reverse step

        $Z \sim \mathrm{N}(0, \mathrm{Id})$      ▷ Sample tangent noise

        $\tilde{\boldsymbol{y}}_t^* = \tilde{\boldsymbol{y}}_{t-\gamma}^* + \gamma \left\{ \frac{1}{2} \left( m(\boldsymbol{x}^*) - \tilde{\boldsymbol{y}}_{t-\gamma}^* \right) \right\} + \sqrt{\gamma} \mathrm{K}(\boldsymbol{x}^*, \boldsymbol{x}^*)^{1/2} Z$      ▷ Forward step

    $\boldsymbol{y}_{t-\gamma}^* = \tilde{\boldsymbol{y}}_{t-\gamma}^*$

**return** $\boldsymbol{y}_\epsilon^*$

---

First, we start by describing the RePaint algorithm (Lugmayr et al., 2022). We consider $(Z_k^0, Z_k^1, Z_k^2)_{k \in \mathbb{N}}$ a sequence of independent Gaussian random variable such that for any $k \in \mathbb{N}$, $Z_k^1$ and $Z_k^2$ are $d$-dimensional Gaussian random variables with zero mean and identity covariance matrix and $Z_k^0$ is a $p$-dimensional Gaussian random variable with zero mean and identity covariance matrix. We assume that the whole sequence to be inferred is of size $d$ while the context is of size $p$. For simplicity, we only consider the Euclidean setting with $\mathrm{K} = \mathrm{Id}$. The proofs can be adapted to cover the case $\mathrm{K} \neq \mathrm{Id}$ without loss of generality.

Let us fix a time $t_0 \in [0, T]$. We consider the chain $(X_k)_{k \in \mathbb{N}}$ given by $X_0 \in \mathbb{R}^d$ and for any $k \in \mathbb{N}$, we define

$$X_{k+1/2} = \mathrm{e}^\gamma X_k + 2(\mathrm{e}^\gamma - 1) \nabla_{x_k} \log p_{t_0}([X_k, X_k^c]) + (\mathrm{e}^{2\gamma} - 1)^{1/2} Z_k^1, \tag{69}$$

where $X_k^c = \mathrm{e}^{-t_0} X_0^c + (1 - \mathrm{e}^{-2t_0})^{1/2} Z_k^0$. Finally, we consider

$$X_{k+1} = \mathrm{e}^{-\gamma} X_{k+1/2} + (1 - \mathrm{e}^{-2\gamma})^{1/2} Z_k^2. \tag{70}$$

Note that (69) corresponds to one step of *backward SDE* integration and (70) corresponds to one step of *forward SDE* integration. In both cases we have used the exponential integrator, see (De

Bortoli, 2022) for instance. While we use the exponential integrator in the proofs for simplicity other integrators such as the classical Euler-Maruyama integration could have been used. Combining (69) and (70), we get that for any $k \in \mathbb{N}$ we have

$$X_{k+1} = X_k + 2(1 - \mathrm{e}^{-\gamma})\nabla_{x_k} \log p_{t_0}([X_k, X_k^c]) + (1 - \mathrm{e}^{-2\gamma})^{1/2}(Z_k^1 + Z_k^2). \tag{71}$$

Remarking that $(Z_k)_{k\in\mathbb{N}} = ((Z_k^1 + Z_k^3)/\sqrt{2})_{k\in\mathbb{N}}$ is a family of $d$-dimensional Gaussian random variables with zero mean and identity covariance matrix, we get that for any $k \in \mathbb{N}$

$$X_{k+1} = X_k + 2(1 - \mathrm{e}^{-\gamma})\nabla_{x_k} \log p_{t_0}([X_k, X_k^c]) + \sqrt{2}(1 - \mathrm{e}^{-2\gamma})^{1/2}Z_k, \tag{72}$$

where we recall that $X_k^c = \mathrm{e}^{-t_0}X_0^c + (1 - \mathrm{e}^{-2t_0})^{1/2}Z_k^0$. Note that the process (72) is another version of the `Repaint` algorithm (Lugmayr et al., 2022), where we have concatenated the denoising and noising procedure. With this formulation, it is clear that `Repaint` is equivalent to Alg. 1. In what follows, we identify the limiting SDE of this process.

In what follows, we describe the limiting behavior of (72) under mild assumptions on the target distribution. In what follows, for any $x_{t_0} \in \mathbb{R}^d$, we denote

$$b(x_{t_0}) = 2 \int_{\mathbb{R}^p} \nabla_{x_{t_0}} \log p_{t_0}([x_{t_0}, x_{t_0}^c])p_{t_0|0}(x_{t_0}^c|x_0^c)\mathrm{d}x_{t_0}^c. \tag{73}$$

We emphasize that $b/2 \neq \nabla_{x_{t_0}} \log p(\cdot|x_0^c)$. In particular, using Tweedie's identity, we have that for any $x_{t_0} \in \mathbb{R}^d$

$$\nabla \log p_{t_0}(x_{t_0}|x_0^c) = \int_{\mathbb{R}^p} \nabla_{x_{t_0}} \log p([x_{t_0}, x_{t_0}^c]|x_0^c)p(x_{t_0}^c|x_{t_0}, x_0^c)\mathrm{d}x_{t_0}^c. \tag{74}$$

We introduce the following assumption.

**Assumption 1.** *There exist* $\mathrm{L}, C \geq 0$, $\mathrm{m} > 0$ *such that for any* $x_{t_0}^c, y_t^c \in \mathbb{R}^p$ *and* $x_{t_0}, y_t \in \mathbb{R}^d$

$$\|\nabla \log p_{t_0}([x_{t_0}, x_{t_0}^c]) - \nabla \log p_{t_0}([y_t, y_t^c])\| \leq \mathrm{L}(\|x_{t_0} - y_t\| + \|x_{t_0}^c - y_t^c\|). \tag{75}$$

Assumption 1 ensures that there exists a unique strong solution to the SDE associated with (72). Note that conditions under which $\log p_{t_0}$ is Lipschitz are studied in De Bortoli (2022). In the theoretical literature on diffusion models the Lipschitzness assumption is classical, see Lee et al. (2023) and Chen et al. (2022).

We denote $((\mathbf{X}_t^\gamma)_{t\geq 0})_{\gamma > 0}$ the family of processes such that for any $k \in \mathbb{N}$ and $\gamma > 0$, we have for any $t \in [k\gamma, (k+1)\gamma)$, $\mathbf{X}_t^\gamma = (1 - (t - k\gamma)/\gamma)\mathbf{X}_{k\gamma}^\gamma + (t - k\gamma)/\gamma\mathbf{X}_{(k+1)\gamma}^\gamma$ and

$$\mathbf{X}_{(k+1)\gamma}^\gamma = \mathbf{X}_{k\gamma}^\gamma + 2(1 - \mathrm{e}^{-\gamma})\nabla_{\mathbf{X}_{k\gamma}^\gamma} \log p_{t_0}([\mathbf{X}_{k\gamma}^\gamma, \mathbf{X}_{k\gamma}^{c,n}]) + \sqrt{2}(1 - \mathrm{e}^{-2\gamma})^{1/2}\mathbf{Z}_{k\gamma}^\gamma, \tag{76}$$

where $(\mathbf{Z}_{k\gamma}^\gamma)_{k\in\mathbb{N}, \gamma > 0}$ is a family of independent $d$-dimensional Gaussian random variables with zero mean and identity covariance matrix and for any $k \in \mathbb{N}$, $\gamma > 0$, $\mathbf{X}_{k\gamma}^{c,\gamma} = \mathrm{e}^{-t_0}x_0^c + (1 - \mathrm{e}^{-2t_0})^{1/2}\mathbf{Z}_{k\gamma}^{0,\gamma}$, where $(\mathbf{Z}_{k\gamma}^{0,\gamma})_{k\in\mathbb{N}, \gamma > 0}$ is a family of independent $p$-dimensional Gaussian random variables with zero mean and identity covariance matrix. This is a *linear interpolation* of the `Repaint` algorithm in the form of (72).

Finally, we denote $(\mathbf{X}_t)_{t\geq 0}$ such that

$$\mathrm{d}\mathbf{X}_t = b(\mathbf{X}_t)\mathrm{d}t + 2\mathbf{B}_t, \qquad \mathbf{X}_0 = x_0. \tag{77}$$

We recall that $b$ depends on $t_0$ but $t_0$ is *fixed* here. This means that we are at time $t_0$ in the diffusion and consider a *corrector* at this stage. The variable $t$ does not corresponds to the backward evolution but to the forward evolution *in the corrector stage*. Under Assumption 1, (77) admits a unique strong solution. The rest of the section is dedicated to the proof of the following result.

**Theorem E.1.** *Assume* Assumption 1. *Then* $\lim_{n\to+\infty}(\mathbf{X}_t^{1/n})_{t\geq 0} = (\mathbf{X}_t)_{t\geq 0}$.

This result is an application of Stroock and Varadhan (2007, Theorem 11.2.3). It explicits what is the *continuous* limit of the `Repaint` algorithm (Lugmayr et al., 2022).

In what follows, we verify that the assumptions of this result hold in our setting. For any $\gamma > 0$ and $x \in \mathbb{R}^d$, we define

$$b_\gamma(x) = (2/\gamma)[(1 - \mathrm{e}^{-\gamma}) \int_{\mathbb{R}^d} \nabla_{x_{t_0}} \log p_{t_0}([x_{t_0}, x_{t_0}^c]) p_{t_0|0}(x_{t_0}^c | x_0^c) \mathrm{d}x_{t_0}^c \tag{78}$$

$$- (1/\gamma) \mathbb{E}[(\mathbf{X}_{(k+1)\gamma}^\gamma - \mathbf{X}_{k\gamma}^\gamma) \mathbf{1}_{\|\mathbf{X}_{(k+1)\gamma}^\gamma - \mathbf{X}_{k\gamma}^\gamma\| \geq 1} \,|\, \mathbf{X}_{k\gamma} = x], \tag{79}$$

$$\Sigma_\gamma(x) = (4/\gamma)(1 - \mathrm{e}^{-\gamma})^2 \int_{\mathbb{R}^d} \nabla_{x_{t_0}} \log p_{t_0}([x_{t_0}, x_{t_0}^c])^{\otimes 2} p_{t_0|0}(x_{t_0}^c | x_0^c) \mathrm{d}x_{t_0}^c + (2/\gamma)(1 - \mathrm{e}^{-2\gamma}) \,\mathrm{Id} \tag{80}$$

$$- (1/\gamma) \mathbb{E}[(\mathbf{X}_{(k+1)\gamma}^\gamma - \mathbf{X}_{k\gamma}^\gamma)^{\otimes 2} \mathbf{1}_{\|\mathbf{X}_{(k+1)\gamma}^\gamma - \mathbf{X}_{k\gamma}^\gamma\| \geq 1} \,|\, \mathbf{X}_{k\gamma} = x]. \tag{81}$$

Note that for any $\gamma > 0$ and $x \in \mathbb{R}^d$, we have

$$b_\gamma(x) = \mathbb{E}[\mathbf{1}_{\|\mathbf{X}_{(k+1)\gamma}^\gamma - \mathbf{X}_{k\gamma}^\gamma\| \leq 1}(\mathbf{X}_{(k+1)\gamma}^\gamma - \mathbf{X}_{k\gamma}^\gamma) \,|\, \mathbf{X}_{k\gamma}^\gamma = x] \tag{82}$$

$$\Sigma_\gamma(x) = \mathbb{E}[\mathbf{1}_{\|\mathbf{X}_{(k+1)\gamma}^\gamma - \mathbf{X}_{k\gamma}^\gamma\| \leq 1}(\mathbf{X}_{(k+1)\gamma}^\gamma - \mathbf{X}_{k\gamma}^\gamma)^{\otimes 2} \,|\, \mathbf{X}_{k\gamma}^\gamma = x] \tag{83}$$

$$\tag{84}$$

**Lemma E.2.** *Assume* Assumption 1. *Then, we have that for any $R, \varepsilon > 0$ and $\gamma \in (0, 1)$*

$$\lim_{\gamma \to 0} \sup\{\|\Sigma_\gamma(x) - \Sigma(x)\| \,|\, x \in \mathbb{R}^d, \ \|x\| \leq R\} = 0, \tag{85}$$

$$\lim_{\gamma \to 0} \sup\{\|b_\gamma(x) - b(x)\| \,|\, x \in \mathbb{R}^d, \ \|x\| \leq R\} = 0, \tag{86}$$

$$\lim_{\gamma \to 0} (1/\gamma) \sup\{\mathbb{P}(\|\mathbf{X}_{(k+1)\gamma}^\gamma - \mathbf{X}_{k\gamma}^\gamma\| \geq \varepsilon \,|\, \mathbf{X}_{k\gamma} = x) \,|\, x \in \mathbb{R}^d, \ \|x\| \leq R\} = 0. \tag{87}$$

*Where we recall that for any $x \in \mathbb{R}^d$,*

$$b(x) = 2 \int_{\mathbb{R}^p} \nabla_{x_{t_0}} \log p_{t_0}([x_{t_0}, x_{t_0}^c]) p_{t|0}^x(x_{t_0}^c | x_0^c) \mathrm{d}x_{t_0}^c, \qquad \Sigma(x) = 4 \,\mathrm{Id}. \tag{88}$$

*Proof.* Let $R, \varepsilon > 0$ and $\gamma \in (0, 1)$. Using Assumption 1, there exists $C > 0$ such that for any $x_{t_0} \in \mathbb{R}^d$ with $\|x_{t_0}\| \leq R$, we have $\|\nabla_{x_{t_0}} \log p_{t_0}([x_{t_0}, x_{t_0}^c])\| \leq C(1 + \|x_{t_0}^c\|)$. Since $p_{t_0|0}^c$ is Gaussian with zero mean and covariance matrix $(1 - \mathrm{e}^{-2t_0}) \,\mathrm{Id}$, we get that for any $p \in \mathbb{N}$, there exists $A_k \geq 0$ such that for any $x_{t_0} \in \mathbb{R}^d$ with $\|x_{t_0}\| \leq R$

$$\int_{\mathbb{R}^d} \|\nabla_{x_{t_0}} \log p_{t_0}([x_{t_0}, x_{t_0}^c])\|^p p_{t_0|0}^c(x_{t_0}^c | x_0^c) \mathrm{d}x_{t_0}^c \leq A_k(1 + \|x_0^c\|^p). \tag{89}$$

Therefore, using this result and the fact that for any $s \geq 0$, $\mathrm{e}^{-s} \geq 1 - s$, we get that there exists $B_k \geq 0$ such that for any $k, p \in \mathbb{N}$ and for any $x_{t_0} \in \mathbb{R}^d$ with $\|x_{t_0}\| \leq R$

$$\mathbb{E}[\|\mathbf{X}_{(k+1)\gamma} - \mathbf{X}_{k\gamma}\|^p \,|\, \mathbf{X}_{k\gamma} = x] \leq B_k \gamma^{p/2}(1 + \|x_0^c\|^p). \tag{90}$$

Therefore, combining this result and the Markov inequality, we get that for any $x_{t_0} \in \mathbb{R}^d$ with $\|x_{t_0}\| \leq R$ we have

$$\lim_{\gamma \to 0}(1/\gamma) \sup\{\mathbb{P}(\|\mathbf{X}_{(k+1)\gamma}^\gamma - \mathbf{X}_{k\gamma}^\gamma\| \geq \varepsilon \,|\, \mathbf{X}_{k\gamma} = x) \,|\, x \in \mathbb{R}^d, \ \|x\| \leq R\} = 0, \tag{91}$$

$$\lim_{\gamma \to 0}(1/\gamma)\|\mathbb{E}[(\mathbf{X}_{(k+1)\gamma}^\gamma - \mathbf{X}_{k\gamma}^\gamma) \mathbf{1}_{\|\mathbf{X}_{(k+1)\gamma}^\gamma - \mathbf{X}_{k\gamma}^\gamma\| \geq 1} \,|\, \mathbf{X}_{k\gamma} = x]\| = 0, \tag{92}$$

$$\lim_{\gamma \to 0}(1/\gamma)\|\mathbb{E}[(\mathbf{X}_{(k+1)\gamma}^\gamma - \mathbf{X}_{k\gamma}^\gamma) \mathbf{1}_{\|\mathbf{X}_{(k+1)\gamma}^\gamma - \mathbf{X}_{k\gamma}^\gamma\| \geq 1} \,|\, \mathbf{X}_{k\gamma} = x]\| = 0 \tag{93}$$

In addition, we have that for any $x_{t_0} \in \mathbb{R}^d$ with $R > 0$

$$|(2/\gamma)(1 - \mathrm{e}^{-\gamma}) - 2|\|\int_{\mathbb{R}^d} \nabla_{x_{t_0}} \log p_{t_0}([x_{t_0}, x_{t_0}^c]) p_{t_0|0}(x_{t_0}^c | x_0^c) \mathrm{d}x_{t_0}^c\| \tag{94}$$

$$\leq A_1(1 + \|x_0^c\|)(2/\gamma)|\mathrm{e}^{-\gamma} - 1 + \gamma|. \tag{95}$$

We also have that for any $x_{t_0} \in \mathbb{R}^d$ with $R > 0$

$$(4/\gamma)|1 - \mathrm{e}^{-\gamma}|^2 \|\int_{\mathbb{R}^d} \nabla_{x_{t_0}} \log p_{t_0}([x_{t_0}, x_{t_0}^c])^{\otimes 2} p_{t_0|0}(x_{t_0}^c | x_0^c) \mathrm{d}x_{t_0}^c\| \tag{96}$$

$$\leq A_2(1 + \|x_0^c\|^2)(4/\gamma)|1 - \mathrm{e}^{-\gamma}|^2. \tag{97}$$

Combining this result, (91), the fact that $\lim_{\gamma \to 0}(4/\gamma)|1 - \mathrm{e}^{-\gamma}|^2 = 0$ and $\lim_{\gamma \to 0}(2/\gamma)|\mathrm{e}^{-\gamma} - 1 + \gamma| = 0$, we get that $\lim_{\gamma \to 0} \sup\{\|\Sigma_\gamma(x) - \Sigma(x)\| \,|\, x \in \mathbb{R}^d, \ \|x\| \leq R\} = 0$. Similarly, using (91), (94) and the fact that $\lim_{\gamma \to 0}(4/\gamma)|1 - \mathrm{e}^{-\gamma}|^2 = 0$, we get that $\lim_{\gamma \to 0} \sup\{\|b_\gamma(x) - b(x)\| \,|\, x \in \mathbb{R}^d, \ \|x\| \leq R\} = 0$. $\square$

We can now conclude the proof of Theorem E.1.

*Proof.* We have that $x \mapsto b(x)$ and $x \mapsto \Sigma(x)$ are continuous. Combining this result and Lemma E.2, we conclude the proof upon applying Stroock and Varadhan (2007, Theorem 11.2.3). $\qquad\square$

Theorem E.1 is a non-quantitative result which states what is the limit chain for the REPAINT procedure. Note that if we do not resample, we get that

$$b^{\mathrm{cond}}(x) = 2\nabla_{x_{t_0}} \log p_{t_0}([x_{t_0}, x_{t_0}^c]), \qquad \Sigma(x) = 4\,\mathrm{Id}. \tag{98}$$

Recalling (88), we get that (98) is an *amortised version* of $b^{\mathrm{cond}}$. Similar convergence results can be derived in this case. Note that it is also possible to obtain quantitative discretization bounds between $(\mathbf{X}_t)_{t \geq 0}$ and $(\mathbf{X}_t^{1/n})_{t \geq 0}$ under the $\ell^2$ distance. These bounds are usually leveraged using the Girsanov theorem (Durmus and Moulines, 2017; Dalalyan, 2017). We leave the study of such bounds for future work.

We also remark that $b(x_{t_0})$ is *not* given by $\nabla \log p_{t_0}(x_{t_0}|x_0^c)$. Denoting $U_{t_0}$ such that for any $x_{t_0} \in \mathbb{R}^d$

$$U_{t_0}(x_{t_0}) = - \int_{\mathbb{R}^p} (\log p_{t_0}(x_{t_0}|x_{t_0}^c)) p_{t|0}(x_{t_0}^c|x_0^c) \mathrm{d}x_{t_0}^c, \tag{99}$$

we have that $\nabla U_{t_0}(x_{t_0}) = -b(x_{t_0})$, under mild integration assumptions. In addition, using Jensen's inequality, we have

$$\int_{\mathbb{R}^d} \exp[-U_{t_0}(x_{t_0})]\mathrm{d}x_{t_0} \leq \int_{\mathbb{R}^d} \int_{\mathbb{R}^p} p_{t_0}(x_{t_0}|x_{t_0}^c) p_{t|0}(x_{t_0}^c|x_0^c)\mathrm{d}x_{t_0}\mathrm{d}x_{t_0}^c \leq 1. \tag{100}$$

Hence, $\pi_{t_0}$ with density proportional to $x \mapsto \exp[-U_{t_0}(x)]$ defines a valid probability measure.

We make the following assumption which allows us to control the ergodicity of the process $(\mathbf{X}_t)_{t \geq 0}$.

**Assumption 2.** *There exist* $\mathtt{m} > 0$ *and* $C \geq 0$ *such that for any* $x_{t_0} \in \mathbb{R}^d$ *and* $x_{t_0}^c \in \mathbb{R}^p$

$$\langle \nabla_{x_t} \log p_{t_0}([x_t, x_t^c]), x_t \rangle \leq -\mathtt{m}\|x_t\|^2 + C(1 + \|x_t^c\|^2). \tag{101}$$

The following proposition ensures the ergodicity of the chain $(\mathbf{X}_t)_{t \geq 0}$. It is a direct application of Roberts and Tweedie (1996, Theorem 2.1).

**Proposition E.3.** *Assume* Assumption 1 *and* Assumption 2. *Then,* $\pi_{t_0}$ *is the unique invariant probability measure of* $(\mathbf{X}_t)_{t \geq 0}$ *and* $\lim_{t \to 0} \|\mathcal{L}(\mathbf{X}_t) - \pi_{t_0}\|_{\mathrm{TV}} = 0$, *where* $\mathcal{L}(\mathbf{X}_t)$ *is the distribution of* $\mathbf{X}_t$.

Finally, for any $t_0 > 0$, denoting $\pi_{t_0}$ the probability measure with density $U_{t_0}$ given for any $x_{t_0} \in \mathbb{R}^d$ by

$$U_{t_0}(x_{t_0}) = - \int_{\mathbb{R}^p} (\log p_{t_0}(x_{t_0}|x_{t_0}^c)) p_{t|0}(x_{t_0}^c|x_0^c)\mathrm{d}x_{t_0}^c. \tag{102}$$

We show that the family of measures $(\pi_{t_0})_{t_0 > 0}$ approximates the posterior with density $x_0 \mapsto p(x_0|x_0^c)$ when $t_0$ is small enough.

**Proposition E.4.** *Assume* Assumption 1. *We have that* $\lim_{t_0 \to 0} \pi_{t_0} = \pi_0$ *where* $\pi_0$ *admits a density w.r.t. the Lebesgue measure given by* $x_0 \mapsto p(x_0|x_0^c)$.

*Proof.* This is a direct consequence of the fact that $p_{t|0}(\cdot|x_0^c) \to \delta_{x_0^c}$. $\qquad\square$

This last results shows that even though we do not target $x_{t_0} \mapsto p_{t_0|0}(x_{t_0}|x_0^c)$ using this corrector term, we still target $p(x_0|x_0^c)$ as $t_0 \to 0$ which corresponds to the desired output of the algorithm.

# F   Experimental details

Models, training and evaluation have been implemented in Jax (Bradbury et al., 2018). We used Python (Van Rossum and Drake Jr, 1995) for all programming, Hydra (Yadan, 2019), Numpy (Harris et al., 2020), Scipy (Virtanen et al., 2020), Matplotlib (Hunter, 2007), and Pandas (McKinney et al., 2010). The code is publicly available at https://github.com/cambridge-mlg/neural_diffusion_processes.

### F.1 Regression 1d

#### F.1.1 Data generation

We follow the same experimental setup as Bruinsma et al. (2020) to generate the 1d synthetic data. It consists of Gaussian (Squared Exponential (SE), MATÉRN($\frac{5}{2}$), WEAKLY PERIODIC) and non-Gaussian (SAWTOOTH and MIXTURE) sample paths, where MIXTURE is a combination of the other four datasets with equal weight. Fig. 9 shows samples for each of these dataset. The Gaussian datasets are corrupted with observation noise with variance $\sigma^2 = 0.05^2$. The left column of Fig. 9 shows example sample paths for each of the 5 datasets.

The training data consists of $2^{14}$ sample paths while the test dataset has $2^{1}2$ paths. For each test path we sample the number of context points between 1 and 10, the number of target points are fixed to 50 for the GP datasets and 100 for the non-Gaussian datasets. The input range for the training and interpolation datasets is $[-2, 2]$ for both the context and target sets, while for the extrapolation task the context and target input points are drawn from $[2, 6]$.

**Architecture.** For all datasets, except SAWTOOTH, we use 5 bi-dimensional attention layers (Dutordoir et al., 2022) with 64 hidden dimensions and 8 output heads. For SAWTOOTH, we obtained better performance with a wider and shallower model consisting of 2 bi-dimensional attention layers with a hidden dimensionality of 128. In all experiment, we train the NDP-based models over 300 epochs using a batch size of 256. Furthermore, we use the Adam optimiser for training with the following learning rate schedule: linear warm-up for 10 epochs followed by a cosine decay until the end of training.

#### F.1.2 Ablation Limiting Kernels

The test log-likelihoods (TLLs) reported in App. F.1.3 for the NDP models target a white limiting kernel and train to approximate the preconditioned score $K\nabla \log p_t$. Overall, we found this to be the best performing setting. App. F.1.3 shows an ablation study for different choices of limiting kernel and score parametrisation. We refer to Table 3 for a detailed derivation of the score parametrisations.

The dataset in the top row of the figure originates from a Squared Exponential (SE) GP with lengthscale $\ell = 0.25$. We compare the performance of three different limiting kernels: white (blue), a SE with a longer lengthscale $\ell = 1$ (orange), and a SE with a shorter lengthscale $\ell = 0.1$ (green). As the dataset is Gaussian, we have access to the true score. We observe that, across the different parameterisations, the white limiting kernel performance best. However, note that for the White kernel $K = I$ and thus the different parameterisations become identical. For non-white limiting kernels we see a reduction in performance for both the approximate and exact score. We attribute this to the additional complexity of learning a non-diagonal covariance.

In the bottom row of App. F.1.3 we repeat the experiment for a dataset consisting of samples from the Periodic GP with lengthscale $0.5$. We draw similar conclusions: the best performing limiting kernel, across the different parametrisations, is the White noise kernel.

#### F.1.3 Ablation Conditional Sampling

Next, we focus on the empirical performance of the different noising schemes in the conditional sampling, as discussed in Fig. 8. For this, we measure the the Kullback-Leibler (KL) divergence between two Gaussian distributions: the true GP-based conditional distribution, and an distribution created by drawing conditional sampling from the model and fitting a Gaussian to it using the empirical mean and covariance. We perform this test on the 1D squared exponential dataset (described above) as this gives us access to the true posterior. We use $2^{12}$ samples to estimate the empirical mean and covariance, and fix the number of context points to 3.

In Fig. 11 we keep the total number of score evaluations fixed to 5000 and vary the number of steps in the inner ($L$) loop such that the number of outer steps is given by the ratio $5000/L$. From the figure, we observe that the particular choice of noising scheme is of less importance as long at least a couple ($\pm 5$) inner steps are taken. We further note that in this experiment we used the true score (available because of the Gaussianity of the dataset), which means that these results may differ if an approximate score network is used.

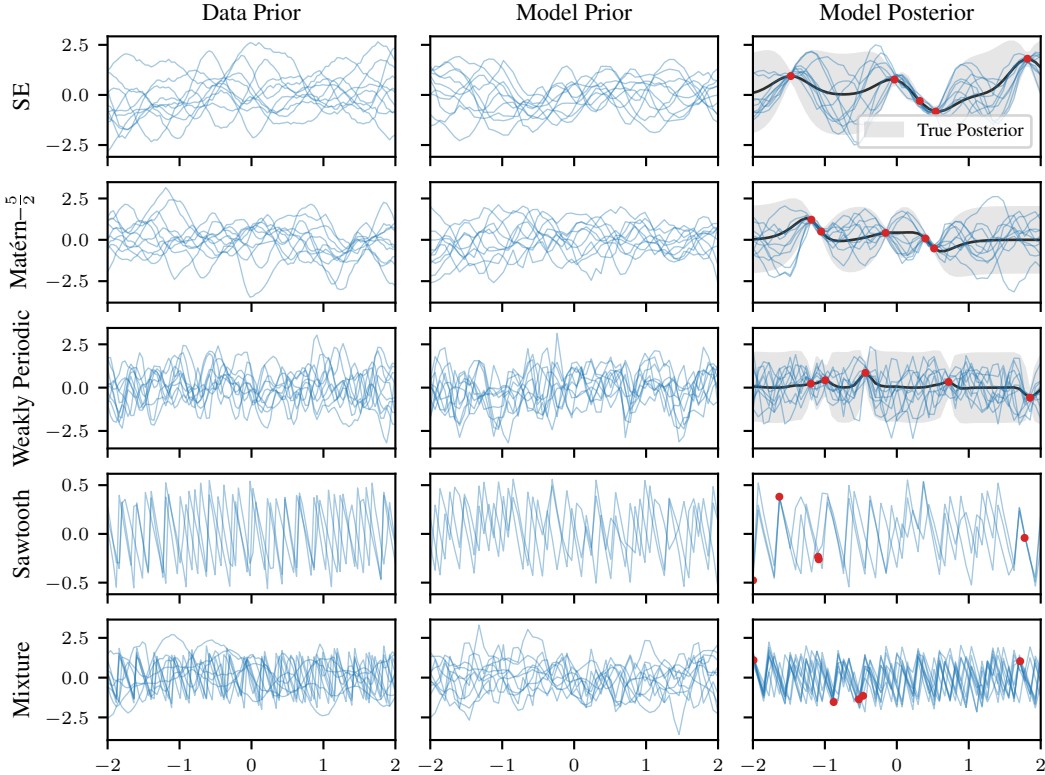

Figure 9: Visualisation of 1D regression experiment.

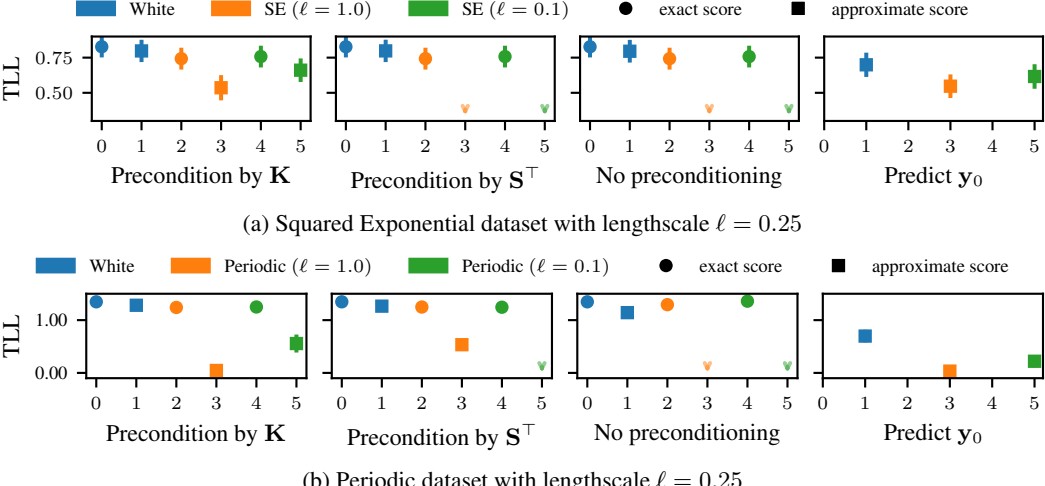

(a) Squared Exponential dataset with lengthscale $\ell = 0.25$

(b) Periodic dataset with lengthscale $\ell = 0.25$

Figure 10: *Ablation study* targeting different limiting kernels and score parametrisations.

Table 5: Mean test log-likelihood (TLL) $\pm$ 1 standard error estimated over 4096 test samples are reported. Statistically significant best non-GP model is in bold. The NP baselines (GNP, ConvCNP, ConvNP and ANP) are quoted from Bruinsma et al. (2020). '*' stands for a TLL below -10.

| | SE | Matérn$-\frac{5}{2}$ | Weakly Per. | Sawtooth | Mixture |
|---|---|---|---|---|---|
| **Interpolation** | | | | | |
| GP (optimum) | $0.70\pm_{0.00}$ | $0.31\pm_{0.00}$ | $-0.32\pm_{0.00}$ | n/a | n/a |
| $T(1)-$GeomNDP | $\mathbf{0.72}\pm_{0.03}$ | $\mathbf{0.32}\pm_{0.03}$ | $\mathbf{-0.38}\pm_{0.03}$ | $\mathbf{3.39}\pm_{0.04}$ | $\mathbf{0.64}\pm_{0.08}$ |
| NDP* | $\mathbf{0.71}\pm_{0.03}$ | $0.30\pm_{0.03}$ | $\mathbf{-0.37}\pm_{0.03}$ | $\mathbf{3.39}\pm_{0.04}$ | $\mathbf{0.64}\pm_{0.08}$ |
| GNP | $\mathbf{0.70}\pm_{0.01}$ | $0.30\pm_{0.01}$ | $-0.47\pm_{0.01}$ | $0.42\pm_{0.01}$ | $0.10\pm_{0.02}$ |
| ConvCNP | $-0.80\pm_{0.01}$ | $-0.95\pm_{0.01}$ | $-1.20\pm_{0.01}$ | $0.55\pm_{0.02}$ | $-0.93\pm_{0.02}$ |
| ConvNP | $-0.46\pm_{0.01}$ | $-0.67\pm_{0.01}$ | $-1.02\pm_{0.01}$ | $1.20\pm_{0.01}$ | $-0.50\pm_{0.02}$ |
| ANP | $-0.61\pm_{0.01}$ | $-0.75\pm_{0.01}$ | $-1.19\pm_{0.01}$ | $0.34\pm_{0.01}$ | $-0.69\pm_{0.02}$ |
| **Generalisation** | | | | | |
| GP (optimum) | $0.70\pm_{0.00}$ | $0.31\pm_{0.00}$ | $-0.32\pm_{0.00}$ | n/a | n/a |
| $T(1)-$GeomNDP | $\mathbf{0.70}\pm_{0.02}$ | $\mathbf{0.31}\pm_{0.02}$ | $\mathbf{-0.38}\pm_{0.03}$ | $\mathbf{3.39}\pm_{0.03}$ | $\mathbf{0.62}\pm_{0.02}$ |
| NDP* | * | * | * | * | * |
| GNP | $\mathbf{0.69}\pm_{0.01}$ | $\mathbf{0.30}\pm_{0.01}$ | $-0.47\pm_{0.01}$ | $0.42\pm_{0.01}$ | $0.10\pm_{0.02}$ |
| ConvCNP | $-0.81\pm_{0.01}$ | $-0.95\pm_{0.01}$ | $-1.20\pm_{0.01}$ | $0.53\pm_{0.02}$ | $-0.96\pm_{0.02}$ |
| ConvNP | $-0.46\pm_{0.01}$ | $-0.67\pm_{0.01}$ | $-1.02\pm_{0.01}$ | $1.19\pm_{0.01}$ | $-0.53\pm_{0.02}$ |
| ANP | $-1.42\pm_{0.01}$ | $-1.34\pm_{0.01}$ | $-1.33\pm_{0.00}$ | $-0.17\pm_{0.00}$ | $-1.24\pm_{0.01}$ |

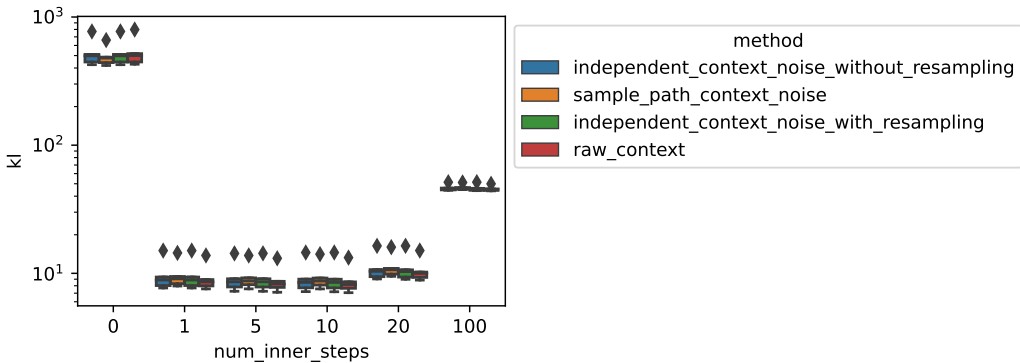

Figure 11: Ablation noising schemes for conditional sampling.

### F.2 Gaussian process vector fields

**Data** We create synthetic datasets using samples from two-dimensional zero-mean GPs with the following E(2)-equivariant kernels: a diagonal Squared-Exponential (SE) kernel, a zero curl (Curl-free) kernel and a zero divergence (Div-free) kernel, as described in App. D.1. We set the variance to $\sigma^2 = 1$ and the lengthscale to $\ell = \sqrt{5}$. We evaluate these GPs on a disk grid, created via a 2D grid with $30 \times 30$ points regularly space on $[-10, 10]^2$ and keeping only the points inside the disk of radius 10. We create a training dataset of size $80 \times 10^3$. and a test dataset of size $10 \times 10^3$.

**Models** We compare two flavours of our model GeomNDP. One with a non-equivariant attention-based score network (Figure C.1, Dutordoir et al., 2022), referred as NDP*. Another one with a E(2)-equivariant score architecture, based on steerable CNNs (Thomas et al., 2018; Weiler et al., 2018). We rely on the e3nn library (Geiger and Smidt, 2022) for implementation. A knn graph $\mathcal{E}$ is built with $k = 20$. The pairwise distances are first embed into $\mu(r_{ab})$ with a 'smooth_finite' basis of 50 elements via e3nn.soft_one_hot_linspace, and with a maximum radius of 2. The time is mapped via a sinusoidal embedding $\phi(t)$ (Vaswani et al., 2017). Then edge features are obtained as $e_{ab} = \Psi^{(e)}(\mu(r_{ab})||\phi(t)) \ \forall(a,b) \in \mathcal{E}_k$ with $\Psi^{(e)}$ an MLP with 2 hidden layers of width 64. We use 5 e3nn.FullyConnectedTensorProduct layers with update given by $V_a^{k+1} = \sum_{b\in\mathcal{N}(a,\mathcal{E}_k)} V_a^k \otimes \left(\Psi^v(e_{ab}||V_a^k||V_b^k)\right) Y(\hat{r}_{ab})$ with $Y$ spherical harmonics up to order 2m $\Psi^v$ an

MLP with 2 hidden layers of width $64$ acting on invariant features, and node features $V^k$ having irreps `12x0e + 12x0o + 4x1e + 4x1o`. Each layer has a gate non-linearity (Weiler et al., 2018).

We also evaluate two neural processes, a translation-equivariant CONVCNP (Gordon et al., 2020) with decoder architecture based on 2D convolutional layers (LeCun et al., 1998) and a $C4 \ltimes \mathbb{R}^2 \subset \mathrm{E}(2)$-equivariant STEERCNP (Holderrieth et al., 2021) with decoder architecture based on 2D steerable convolutions (Weiler and Cesa, 2021). Specific details can be found in the accompanying codebase `https://github.com/PeterHolderrieth/Steerable_CNPs` of Holderrieth et al. (2021).

**Optimisation.** Models are trained for $80k$ iterations, via (Kingma and Ba, 2015) with a learning rate of $5e - 4$ and a batch size of $32$. The neural diffusion processes are trained unconditionally, that is we feed GP samples evaluated on the full disk grid. Their weights are updated via with exponential moving average, with coefficient $0.99$. The diffusion coefficient is weighted by $\beta : t \mapsto \beta_{\min} + (\beta_{\max} - \beta_{\min}) \cdot t$, and $\beta_{\min} = 1e - 4$, $\beta_{\max} = 15$.

As standard, the neural processes are trained by splitting the training batches into a context and evaluation set, similar to when evaluating the models. Models have been trained on A100-SXM-80GB GPUs.

**Evaluation.** We measure the predictive log-likelihood of the data process samples under the model on a held-out test dataset. The context sets are of size 25 and uniformly sampled from a disk grid of size 648, and the models are evaluated on the complementary of the grid. For neural diffusion processes, we estimate the likelihood by solving the associated probability flow ODE (53). The divergence is estimated with the Hutchinson estimator, with Rademacher noise, and 8 samples, whilst the ODE is solved with the 2nd order Heun solver, with 100 discretisation steps.

We also report the performance of the data-generating GP, and the same GP but with diagonal posterior covariance GP (DIAG.).

### F.3 Tropical cyclone trajectory prediction

**Data.** The data is drawn from he International Best330 Track Archive for Climate Stewardship (IBTrACS) Project, Version 4 (Knapp et al., 2010; Knapp et al., 2018). The tracks are taken from the 'all' dataset covering the tracks from all cyclone basins across the globe. The tracks are logged at intervals of every 3 hours. From the dataset, we selected tracks of at least 50 time points long and clipped any longer to this length, resulting in 5224 cyclones. 90% was used fro training and 10% held out for evaluation. This split was changed across seeds. More interesting schemes of variable-length tracks or of interest, but not pursued here in this demonstrative experiment. Natively the track locations live in latitude-longitude coordinates, although it is processed into different forms for different models. The time stamps are processed into the number of days into the cyclone forming and this format is used commonly between all models.

**Models.**

Four models were evaluated.

The GP ($\mathbb{R} \to \mathbb{R}^2$) took the raw latitude-longitude data and normalised it. Using a 2-output RBF kernel with no covariance between the latitude and longitude and taking the cyclone time as input, placed a GP over the data. The hyperparameters of this kernel were optimised using a maximum likelihood grid search over the data. Note that this model places density outside the bounding box of $[-90, 90] \times [-180, 180]$ that defines the range of latitude and longitude, and so does not place a proper distribution on the space of paths on the sphere.

The STEREOGRAPHIC GP ($\mathbb{R} \to \mathbb{R}^2 / \{0\}$) instead transformed the data under a sterographicc projection centred at the north pole, and used the same GP and optimisation as above. Since this model only places density on a set of measure zero that does not correspond to the sphere, it does induce a proper distribution on the space of paths on the sphere.

The NDP ($\mathbb{R} \to \mathbb{R}^2$) uses the same preprocessing as GP ($\mathbb{R} \to \mathbb{R}^2$) but uses a Neural Diffusion Process from (Dutordoir et al., 2022) to model the data. This has the same shortcomings as the GP ($\mathbb{R} \to \mathbb{R}^2$) in not placing a proper density on the space of paths on the sphere. The network used for the score function and the optimisation procedure is detailed below. A linear beta schedule was used with $\beta_0 = 1e - 4$ and $\beta_1 = 10$. The reverse model was integrated back to $\epsilon = 5e - 4$ for numerical

stability. The reference measure was a white noise kernel with a variance $0.05$. ODEs and SDEs were discretised with 1000 steps.

The GEOMNDP ($\mathbb{R} \to \mathcal{S}^2$) works with the data projected into 3d space on the surface of the sphere. This projection makes no difference to the results of the model, but makes the computation of the manifold functions such as the exp map easier, and makes it easier to define a smooth score function on the sphere. This is done by outputting a vector for the score from the neural network in 3d space, and projecting it onto the tangent space of the sphere at the given point. For the necessity of this, see (De Bortoli et al., 2021). The network used for the score function and the optimisation procedure is detailed below. A linear beta schedule was used with $\beta_0 = 1e-4$ and $\beta_1 = 15$. The reverse model was integrated back to $\epsilon = 5e-4$ for numerical stability. The reference measure was a white noise kernel with a variance $0.05$. ODEs and SDEs were discretised with 1000 steps.

**Neural network.** The network used to learn the score function for both NDP ($\mathbb{R} \to \mathbb{R}^2$) and GEOMNDP ($\mathbb{R} \to \mathcal{S}^2$) is a bi-attention network from Dutordoir et al. (2022) with 5 layers, hidden size of 128 and 4 heads per layer. This results in 924k parameters.

**Optimisation.** NDP ($\mathbb{R} \to \mathbb{R}^2$) and GEOMNDP ($\mathbb{R} \to \mathcal{S}^2$) were both optimised using (correctly implemented) Adam for 250k steps using a batch size of 1024 and global norm clipping of 1. Batches were drawn from the shuffled data and refreshed each time the dataset was exhausted. A learning rate schedule was used with 1000 warmup steps linearly from 1e-5 to 1e-3, and from there a cosine schedule decaying from 1e-3 to 1e-5. With even probability either the whole cyclone track was used in the batch, or 20 random points were sub-sampled to train the model better for the conditional sampling task.

**Conditional sampling.** The GP models used closed-form conditional sampling as described. Both diffusion-based models used the Langevin sampling scheme described in this work. 1000 outer steps were used with 25 inner steps. We use a $\psi = 1.0$ and $\lambda_0 = 2.5$. In addition at the end of the Langevin sampling, we run an additional 150 Langevin steps with $t = \epsilon$ as this visually improved performance.

**Evaluation.** For the model (conditional) log probabilities the GP models were computed in closed form. For the diffusion-based models, they were computed using the auxiliary likelihood ODE discretised over 1000 steps. The conditional probabilities were computed via the difference between the log-likelihood of the whole trajectory and the log-likelihood of the context set only. The mean squared errors were computed using the geodesic distance between 10 conditionally sampled trajectories, described above.

