# OpenReview forum: "Geometric Neural Diffusion Processes"
_NeurIPS.cc/2023/Conference — NeurIPS 2023 poster_

### Official Review · Reviewer_fmuk · 2023-07-05

**Soundness:** 3 good
**Presentation:** 3 good
**Contribution:** 3 good
**Rating:** 6
**Confidence:** 4

**Summary:**

This paper proposes several extensions to functional diffusion models. In particular, the focus is on extending the process to cover various geometric inputs (such as manifold values and symmetries), although there are some other components (such as a different formulation of the diffusion that allows for more general kernels and consistency).

**Strengths:**

* The proposed methodology presents a good extension of functional diffusion models. In particular, the more general formulation allows for geometric data, which can be useful for modeling Earth science data. Furthermore, the experiments show that this and the equivariance construction are useful when the datasets exhibit the natural conditions.

**Weaknesses:**

* The main weakness is the experiments. The results in table 1 imply that the alternative diffusion formulation doesn't produce a tangible benefit over the previous formulation when applied to standard data. Furthermore, the other experiments are generally quite toy, which makes it hard to ascertain certain limitations of the method (for example, for S^2 and manifolds in general the kernel is not given in closed form, which can make it hard to extend this methodology to high dimensions).

**Questions:**

N/A

**Limitations:**

Yes

---

> ### Author Rebuttal · Authors · 2023-08-08
>
> We appreciate your thoughtful feedback and the positive assessment of our manuscript.
>
> **Experimental benefits over the previous formulation**
> The goal of Table 1 (bottom part) is to highlight a key advantage of our geometric approach to neural diffusion processes (NDP): the ability to generalize in a data-efficient way to group actions such as roto-translations. This is a feature that previous NDP formulations do not possess. This claim is further evidenced in Figure 5, where our geometric approach requires orders of magnitude less data to obtain superior results compared to the default model. It is also worth noting that the default NDP requires much more parameters (\~1M) than our equivariant model (\~80k), which can further hinder training. Therefore, while the differences between the alternative and previous formulations may not be as pronounced on standard data, it's precisely the capability to generalize and perform efficiently in varied conditions that set the geometric approach apart.
>
> **Toyish experiments**
> Our choice to feature low-dimensional but complex datasets, such as the one-dimensional sawtooth, mixture, and the cyclone dataset on the sphere, was deliberate. These datasets, despite being low-dimensional, include intricate complexities and irregularities.  We believe that this selection underlines not just the robustness of our methodology, but also its versatility when dealing with disparate datasets. While we understand your concerns about potential limitations in extending the methodology to higher dimensions, we feel that our choice of datasets effectively illustrates the strengths of our approach. Finally, with regard to Table 1, we opted to use common datasets from the neural process literature [1,2,3] to empirically assess our model and compare to a range of other neural process-like models.
>
> Thank you for taking the time to review our manuscript. We hope that we have addressed your main concerns in this rebuttal and that you might consider updating your score accordingly.
>
>
> **References**
>
> [1] Foong et al. Meta-Learning Stationary Stochastic Process Prediction with Convolutional Neural Processes. NeurIPS 2020.
>
> [2] Gordon et al. Convolutional conditional Neural Processes. ICLR 2020.
>
> [3] Bruinsma et al. Autoregressive Conditional Neural Processes. ICLR 2023.

---

### Official Review · Reviewer_zVLR · 2023-07-06

**Soundness:** 3 good
**Presentation:** 3 good
**Contribution:** 3 good
**Rating:** 7
**Confidence:** 3

**Summary:**

In this paper, the authors propose a generative model over vector fields. The proposed approach is a combination of ideas from neural diffusion processes [1], equivariant learning of stochastic fields [2] and equivariant scores [3]. The authors represent vector fields as vectors in finite sets of points and use Euclidean groups to incorporate desired symmetries on vector fields. The proposed diffusion model defined via Ornstein–Uhlenbeck process converges to Gaussian Process with equivariant mean and kernel. The model is trained on the denoising score matching loss with score parametrized by an equivariant neural network. This methodology can be extended to manifold-valued functions using Riemannian diffusion models. Further, the authors explore conditional sampling from the proposed model.

[1] Dutordoir, V., Saul, A., Ghahramani, Z., and Simpson, F. (2022). Neural Diffusion Processes.
[2] Holderrieth, P., Hutchinson, M. J., and Teh, Y. W. (2021). Equivariant learning of stochastic fields: Gaussian processes and steerable conditional neural processes. In International Conference on Machine Learning.
[3] Yim, J., Trippe, B. L., De Bortoli, V., Mathieu, E., Doucet, A., Barzilay, R., and Jaakkola, T. (2023). Se (3) diffusion model with application to protein backbone generation. arXiv preprint
arXiv:2302.02277.

**Strengths:**

The paper is well written and provides all necessary information for understanding. Main distinguishing features of proposed model are well demonstrated in experiments. The proposed model needs much less training examples to reach high log-likelihood. In several small-scale experiments the proposed model outperforms existing approaches.

**Weaknesses:**

The experiments are small-dimensional. It would be interesting to see how the proposed approach scales to higher dimensionality, where equivariant models could become even more beneficial.

**Questions:**

Regarding Figure 5: The default NDP fails to achieve the same level of log-likelihood even with large amount of training data. Would it reach the same log-likelihood as E(2)-GeomNDP when the number of training examples tends to “infinity” (> 10^5)? Does it need more parameters? Or is there something else that prevents it from adequately fitting the data?

**Limitations:**

Yes

---

> ### Author Rebuttal · Authors · 2023-08-08
>
> We are grateful for your insightful comments and the favourable evaluation of our manuscript. In the following, we address your primary concerns, aiming to further enhance your appreciation of our work.
>
> **1. Scaling to Higher Dimensionality**
>
> We agree that extending our experiments to higher dimensionalities could indeed yield intriguing results, especially given the potential benefits of equivariant models in such contexts. Your suggestion certainly aligns with our overall aim of comprehensively exploring the potential of the proposed approach. However, such an extension represents a substantial piece of work, likely beyond the scope of this particular paper.
>
> In the current study, we consciously chose to focus on low-dimensional but complex datasets, such as the one-dimensional sawtooth, mixture, and the cyclone dataset on the sphere. These datasets, despite being low-dimensional, include intricate complexities and irregularities. We believe that this selection underlines not just the robustness of our methodology, but also its versatility. Furthermore, the datasets used in Table 1 are the de-facto standard in the neural process (NP) literature (Gordon et al. (2020), Bruinsma et al. (2023), Dutordoir et al. (2023)) allowing us to compare our method to a variety of NP based models.
>
> Regarding scalability, the main computational bottleneck of our method is the number of data points in the dataset due to the quadratic scaling of the attention architecture. We share this constraint with previous models such as Kim et al. (2019) and Dutordoir et al. (2023). Scaling on the input and output dimensions, on the contrary, is linear as a result of the equivariant networks. This makes our method scale favourable with both the input and output dimensions.
>
> **2. Comparison with Default NDP in Figure 5.**
>
> That's indeed a really good remark, we do not believe that more data would help as it appears that the performance is already plateauing. Similarly, we do not believe it comes from the optimisation, as on the contrary equivariant models tend to potentially induce harder optimisation objectives. Hence, it is likely an issue of model capacity. Yet, the non equivariant 'default NDP' already has much more parameters (\~1M) than the equivariant $\mathrm{SE}(3)-\textrm{GeomNDP}$ (\~80k).
>
>
> Again, thank you for taking the time to review our manuscript. We hope that you can share your enthusiasm for our work with the other reviewers in the upcoming discussion period.

---

> > ### Comment · Reviewer_zVLR · 2023-08-14
> >
> > Thank you for clarifying. I would like to keep my score.

---

### Official Review · Reviewer_NoKK · 2023-07-06

**Soundness:** 4 excellent
**Presentation:** 3 good
**Contribution:** 3 good
**Rating:** 7
**Confidence:** 4

**Summary:**

This paper proposes GeomNDP, a continuous-time functional diffusion model which can incorporate symmetries of the problem of interest. In particular, this model is constructed via a collection of finite-dimensional Ornstein-Uhlenbeck processes (one for each finite-dimensional marginal), which target a given prior Gaussian process. In addition, the authors study (Prop. 3.2, 3.3) conditions under which the resulting process is invariant to a given group. A conditional sampling method is also proposed (Section 3.3) which generalizes the RePAINT method (Lugmayr et al.) to the functional setting.

**Strengths:**

- The paper is well-written, clear, and generally has a high-quality presentation.
- This work contributes to the growing literature on functional diffusion models by incorporating group invariances into their design. This is likely to be of interest to the community.
- The proposed model shows competitive performance against several baselines across multiple tasks. In general, the experimental evaluation is clear and convincing.
- The conditional sampling method achieves empirically strong performance (Figure 7), and is an important contribution to this literature. As noted by the authors, previous conditional functional diffusion models use ad-hoc methods which do not target the correct distribution.
- The work is well-contextualized within the relevant literature.

**Weaknesses:**

- The exposition of the conditional sampler (Section 3.3) was somewhat unclear (see questions), and required a careful read of the pseudocode in order to understand what exactly was being proposed.
- It would have been interesting to compare against recently proposed function-space diffusion models (e.g. those discussed in the related work) as additional baselines -- although many of these works are likely to be contemporaneous with the submission, so this is not a major weakness.


### Minor comments
- The submission was not properly anonymized (see the Appendix in the main submission)
- Line 85: it was somewhat unclear what exact assumptions are made on the domain $X$

**Questions:**

- Could you elaborate on the differences between the proposed conditional sampler (Sect. 3.3) and the "replacement" method? E.g. previously proposed conditional functional diffusion models [1-2] work in a manner that, from the description starting on Line 199, in a very similar manner.
- It was unclear to me in what sense (or if at all) defining the noising process on each finite-dimensional marginal corresponds to a well-defined (and reversible) process over the underlying functional measure of interest. Could you provide additional details regarding this correspondence?



[1] Neural Diffusion Processes. Dutordoir, Saul, Ghahramani, Simpson, 2023.

[2] Diffusion Generative Models in Infinite Dimensions. Kerrigan, Ley, Smyth, 2023.


**Limitations:**

The limitations are adequately addressed within the submission.

---

> ### Author Rebuttal · Authors · 2023-08-08
>
> We thank you for your feedback and the positive review. Below, we respond to your main concerns.
>
> **Minor comments:**
> >The submission was not properly anonymized
>
> We apologise for mistakenly attaching an early version of the appendix with the main pdf.
>
> >Line 85: it was somewhat unclear what exact assumptions are made on the domain $X$
>
> The assumptions on $X$ are pretty weak, a metric space would be sufficient for instance. We will update Section 2 to clarify this.
>
> **Questions**
>
> > Clarification on Conditional Sampler
>
> Based on your comment, we have improved the exposition of the conditional sampler in section 3.3. We included the following points:
> - We develop the continuous formulation of the replacement method by Lugmayr et al. (2022), which we theoretically show does not target the correct conditional distribution (as detailed in E.2).
> - We suggest a correction of the algorithm targeting the true conditional distribution using Langevin corrector steps. We theoretically and empirically show this contribution to be crucial for good results.
> - We discuss several initialization schemes for the context variables, which further improve the empirical performance of the conditional sampling algorithm.
>
> >It was unclear to me in what sense (or if at all) defining the noising process on each finite-dimensional marginal corresponds to a well-defined (and reversible) process over the underlying functional measure of interest.
>
> This follows from Kolmogorov Extension Theorem (see, for instance, Øksendal, 2003). One can show that the exchangeability and consistency conditions are satisfied for each finite set of marginals, both in the forward and (true) reverse process. As a result, the finite marginals originate from a stochastic process. We can thus define the (de)noising process on the finite-dimensional marginals without having to explicitly deal with the underlying functional measure of interest. Thank you for the question - we have also clarified this in the manuscript.
>
> Once more, we appreciate the time you've dedicated to reviewing our manuscript. We hope that you convey your enthusiasm for our work to the other reviewers during the forthcoming discussion phase.

---

> > ### Comment · Reviewer_NoKK · 2023-08-14
> >
> > Thank you for the detailed feedback and clarifications. I look forward to reading the updated version of the paper.

---

### Official Review · Reviewer_uLvV · 2023-07-19

**Soundness:** 2 fair
**Presentation:** 3 good
**Contribution:** 3 good
**Rating:** 6
**Confidence:** 3

**Summary:**

The present thesis deals with the problem of learning geometric-sensitive Denoising Diffusion Models (DDMs) for modeling equivariant conditional processes in (non)-Euclidean observation spaces. In doing so, the authors extend existing work that defines diffusions over function spaces with infinite dimensions using a (non-)Euclidean score-based modeling approach. They incorporate a series of geometric priors by constructing a noise process that allows a geometric Gaussian process as the limiting distribution transforming under the symmetry group of interest, and approximate the score as part of the denoising process by a neural network that is equivariant with respect to this group. They show that the generative function model admits the same symmetry under these conditions. The scalability and capacity of the proposed model is evaluated in the context of a novel Langevin-based conditional sampler based on synthetic and real data in the Euclidean and non-Euclidean scenario.

**Strengths:**

I greatly appreciate any attempt to incorporate geometrically sensitive analysis into machine learning. From my perspective it is advantageous, to assume that data incorporates structural information and to give models the ability to address them.

The proposed method builds on a novel combination of well-known techniques, and is original to the best of my knowledge. Further, the paper is quite well positioned to existing work, the content is well organized and, with a few exceptions, has no spelling or grammatical flaws. Empirical evidence is accessed through experiments on toy and real world data and the authors make effort to extend the small-scale experimental setting.

**Weaknesses:**

I would rate the overall added value of the contribution as rather moderate. The biggest shortcomings relate to the clarity of the submission and the technical soundness which I will address with examples in the following.
- Some general points:
  - I'm not very familiar with the theory of Gaussian processes in detail, but I have a solid knowledge of stochastic processes in general and SDEs and Bayesian inference. From this point of view, it is sometimes difficult to keep track of the present work.
  - For me a stochastic process $\mathcal{X}$ is a family of random variables $(X(t,\omega))_{t\in T}$ defined on a common probabillity space $(\Omega, \mathcal{F}, \mathbb{P})$, indexed by some set $T$; for each $t\in T$, $X(t,\bullet):\Omega \to \mathbb{R}^n$ is a measurable function and for $\omega \in \Omega$, $X(\bullet, \omega): T \to \mathbb{R}^{n}$ is a deterministic function, commonly known as path of the process. One can obtain stochastic processes as solutions to stochastic differential equations using the Itô or Stratonovich calculus; a well known example with a closed form solution is the (univariate) Ornstein-Uhlenbeck process, which is also a Gaussian process.
  - Following, e.g., Dutordoir et al. 2022 Sec. 2.1 then a Gaussian process is (univariate) stochastic process $(f(\omega,x))_{x\in\mathbb{R}^D}$ with $f:\Omega \times \mathbb{R}^D \to \mathbb{R}$ such that, for any finite collection of points from $\mathbb{R}^D$ the resulting random vector follows a multivariate normal distribution.
  - In this setting we can model Gaussian processes by using the Kolmogorov extension theorem; but according to Phillips et al. 2022 this comes with difficulties.
  - However if we switch the codomain of the function(s) $f$ and consider $\mathbb{R}^n$ for $n > 1$ we get into trouble how to handle the Gaussian distribution condition for a finite collection of points in this case. Therefore we need a different approach to define Gaussian processes and fortunately the SDE approach turns out to be the right one combined with some convergence results.
  - Last but not least this setting even succeeds in case of non-Euclidean codomain such as compact manifolds.
  - What I want to share with this quick sketch is an example of how I would have liked to have been introduced to the general setting in general. In particular Sec. 3.1 would then have gained in structure.
- Background
  - l. 63 Please provide a convergence result reference.
  - l. 69 Please provide a Can-easily-be-shown result reference.
  - l. 70 Over functions $\mathbf{s}(\mathbf{Y}_t)$ implies $\mathbf{s}:\mathbb{R}^d \to \mathbb{R}^{d}$ or do you mean to minimize over $\mathbf{s}\circ \mathbf{Y}_t$
  - l. 83 Please check wording.

  - l. 73 Can you provide a motivation for considering steerable (feature) fields?
  - l. 89 What exactly is the feature field in (2)?
  - l. 90 What is $p$?
  - l. 93 Can you specify $\rho(h)$ in Fig. 1?
  - l. 94f. I don't understand this comment, can you rephrase it?
- Section 3.1
  - As mentioned at the beginning please highlight the importance of the SDE approach that becomes the right perspective in case of diffusion models for function with codomain $\mathbb{R}^n$ and non-euclidean subspaces
  -  How is a data process defined; can you define the observation model in more detail?
  -  Why is it suddenly important to specify the OU process as multivariate? We already introduced it in Section 2. and what is $\beta_t$ in Eq. (3) and (4)?
  - Please highlight the interpolation approach (Phillips et al. 2022) from your supplementary material Sec. B.1; in my opinion that's a key result at this point. Also please introduce the used convergence results in more detail.
  - l. 120 Do you mean __Gaussian__ white noise?
  - l. 125 Please add a reference for what kind of mild conditions.
  - l. 128 Please add a reference for Stein score.
  - l. 136ff This part (i.e. Training) including the conditional score is unclear to me! Why does the noising process include a score?
  - What numerical solver is used in the manifold case for simulating the reverse diffusion?
- Section 3.2
  - l. 150 Sec. 2 include the definition of a feature field; how is this connected to a tensor field
  - l. 158 (Prop. 3.1) what about the stationary conditional of Theorem 1 (Holderrieth et al.)?
  - l. 165 I was not able to detect the result of Prop. 3.2 in Yim et al. 2023; can you please be more specific?
  - l. 181 Please define or give reference for a definition of a conditional process.
  - l. 184 Please define SP.
- Section 3.3
  - I find it difficult to connected the described method to Alg. 1. Especially, what are detailed reasons why sampling directly at the end time proves difficult. How does Fig. 3 contributes to this section?
  - l. 218 Please check wording?
- Section 4
  - The present work relates to results from De Bortoli et al. 2022, Huang et al. 2022 and Yim et al. 2023 but these are not included into this related work section, why?
- Section 5.1
  - l. 271 Please define GEOMNDP and you are using GeomNDP a few times instead.
  - What about comparing models from Holderrieth et al. and Phillips et al. which are closer conceptional?
  - l. 292 Please define T(1)-GEOMNDP.
  - l. 295 Please define $\mathbb{I}(x=x’)$ or provide reference.
  - l. 305 In my opinion, Fig. 10 is insufficiently evaluated. For example, changing from zero to a single inner step results in a drastic improvement, but then the quality stays nearly the same for 5 and 10 steps; and after that the error increases; Can you elaborate on this observation?

Important side note: Including supplementary material with unfinished notes is not of advantage!

However, I would like to reiterate my appreciation for the authors' interest in expanding the field of neural diffusion models from a geometrical point of view. I'm sure that by working on the deficiencies mentioned, the work will gain in value and will deliver a valuable contribution to the community.

**Questions:**

- Regarding the usage of numerical solvers: From my point of view, numerical (SDE) solvers play a key role in the proposed approach, as it involves the simulation of (reverse) diffusions defined via SDEs. The authors use an Euler-Maruyama discretisation scheme, which is a very naive solver. Despite the fact that more advanced systems would drastically reduce computational efficiency, it would be important to address their benefits. Last but not least, the vanilla Euler-Maruyama scheme is based on a vector space structure. Therefore, in the case of manifold valued SDEs, there are more advanced options that take these intrinsic structures into account, e.g., a geometric Euler-Maruyama scheme. How could this improve the simulation process?
- l. 66 & 72 (4) -> (2) ?
- l. 173 admits -> admit
- l. 179 $A \subset C$ and how is $x^\ast$ defined in Fig. 2?

**Limitations:**

Limitations are addressed (e.g., cf. Chapter 6) and relate to a computational overhead and modeling weaknesses.
No evidence was found that the authors addressed potential negative societal impact. Please make the effort to include this mandatory information in your work.

---

> ### Author Rebuttal · Authors · 2023-08-08
>
> Thank you for taking the time to produce such a thorough review.
>
> In particular, thanks for your concrete suggestion on how to improve section 3.1 - we have updated our manuscript accordingly. Furthermore, due to strong space constraints, we had to limit the number of references and definitions. This may have led to the feeling of a lack of technical soundness. We have incorporated our reply to your comments in the main paper in order to rectify this.
>
> **Background**
>
> - L63: Reference to Bakry et al (2014) and Durmus and Moulines (2015) for Kullback-Leibler and Wasserstein-based convergence results.
> - L69: Reference to Theorem 5.1 of Kallenberg (Foundations of Modern Probability) which can directly be applied.
> - L70: $s:\mathbb{R}^d \rightarrow \mathbb{R}^d$.
> - L73: We added to the "Steerable fields" paragraph that "steerable feature fields" are collections of tensor fields. Cref answer to L150.
> - L89: $g \cdot f(x) = (uh) \cdot f(x) \triangleq \rho(h) f(h^{-1} (x - u))$ is the special case of a $G$-transformed feature field $g \cdot f(x) = \rho(g) f(g^{-1} \cdot x)$ when $G = \mathrm{E}(d) = \mathrm{T}(d) \rtimes \mathrm{O}(d)$.
> - L90: $p$ -> $u$, the translation element of $\mathbb{E}(d)$.
> - L93: We have updated the caption adding that $\rho(h) = h$, with $h$ a 90-degree rotation matrix.
> - L94: Updated to: "For many natural phenomena, a priori we do not want to express a preference for a particular conformation of the feature field and thus want a prior $p$ to place the same density on all the transformed fields: $p(g \cdot f) = p(f), \forall g \in G$."
>
> **Section 3.1**
> - We assume an additive homoscedastic Gaussian observation model and have elaborated on this in the manuscript.
> - You are correct, we will remove "multivariate" to prevent any confusion.
> - The $\beta$-schedule can be seen as rescaling time as $s = \int_{t=0}^s \beta_t \mathrm{d}t$.
> - Phillips et al. 2022: we have highlighted their result in the main text.
> - L125: We reference Haussman & Pardoux (1986) which assume Lipschitz continuity of the drift and volatility matrix and Cattiaux et al. (2021, Theorem 4.9) which only assume a finite entropy condition on the space of processes.
> - L128: Following Song & Ermon (2019) we refer to $\nabla_x \log p_t$ as the _Stein score_.
> - L136: We apologize for the confusion. We have clarified that the noising (forward) process does not include the score.
> - Solver: We rely on the projective solver introduced by Hairer (2011) to enforce the manifold constraint along the trajectory.
>
> **Section 3.2**
> - L150: We apologize for using the terms "tensor field" and "feature field" interchangeably. A tensor field would be defined by the representation $\rho_p(g)=\rho_{\mathrm{triv}}(g)^{\otimes p}$ ($p$ tensor product), e.g. a scalar field would have a type $p=0$, whilst a feature field would refer to a collection of tensors, with possibly a multiplicity of tensors with the same types, and as such would have a representation of the form $\rho = \sum_p a_p \rho_p(g)$.
> - L165: Proposition 3.6 of Yim et al. 2023.
> - L181: We refer to Pollard (A User's Guide to Measure Theoretic Probability, p. 117) for the definition of a conditional process.
> - SP = Stochastic process.
>
> **Section 3.3**
>
> We agree that the sentence "While we could sample directly at the end time this proves difficult in practice" is confusing. This refers to the idea of only using the estimate of the score $s_\theta(t, \cdot)$ at time $t=\epsilon$ with $\epsilon \ll 1$ and then applying Langevin dynamics. This approach fails due to both the highly non-convex nature of $\log p_\epsilon$ (slow mixing) and the fact that $s_\theta(t, \cdot)$ can only be well estimated where $p_\epsilon$ has mass.
> In Alg. 1., the outer loop is the discretisation of the time-reversal process (i.e. predictor step) going from $t \rightarrow t - \gamma$, as illustrated in blue in Fig. 3, while the inner loop is the discretisation of the Langevin dynamics (corrector step) targeting the (conditional) marginal at time $t - \gamma$ as illustrated in red.
>
> **Section 4**
>
> Due to space constraints, we cut part of the paragraph where we were discussing De Bortoli et al., etc., as we believed that referring to them in specific contexts (within Section 3) may be enough, yet we agree that it is actually worth properly discussing.
>
> **Section 5.1**
> - Comparison to Holderrieth et al.
> In the setting where the group of interest is the translation group, then the method of Holderrieth et al. is exactly a convolutional NPs as introduced in Gordon et al.
> - Fig. 10. We discuss this figure in Appendix F.1.3 of the (proper) supplementary material. The _total_ number of predictor and corrector calls is kept fixed, hence the higher number of corrector steps, the smaller the number of predictor steps. It appears that without any Langevin corrector step, the conditional samples are of a poor fit, yet for a high number of corrector steps, the number of predictor steps becomes too small (the discretisation step becomes too large).
>
> **Questions:**
> - Choice of SDE numerical solver: As the time-reversal SDE is not a generic SDE, one can complement the time-reversal step (following a standard SDE solver), with 'corrector' steps. As shown by Song et al. (2019) there is a trade-off between spending computational budget between a higher order predictor (time-reversal steps) and more corrector steps. Regarding the manifold co-domain, indeed there is no vector space structure, we, therefore, leverage geometric random walk (see E. Jørgensen, 1975), which relies on the exponential map to preserve manifold constraint.
> - In fig. 2, $x^*$ is a regular 1D grid for the left-hand side plots, and regular 2D grid restricted to the interior of a disk for the right-hand side plots. We will add this clarification to the caption.
>
> We hope that these clarifications addressed your concerns with regard to the clarity and soundness of the manuscript and that you will be able to update your score accordingly.

---

> > ### Comment · Reviewer_uLvV · 2023-08-15
> >
> > I thank the authors very much for their response and for the detailed discussion
> > of my questions and concerns.
> >
> > You have significantly clarified things, and I will increase my score correspondingly, under
> > the assumption that these clarifications will make it into the updated version
> > of the paper.
> >
> > Please do not shorten these for reasons of space. Otherwise readability will be severely impaired.
> >
> > Thank you again.

---

### Author Rebuttal · Authors · 2023-08-09

Dear Reviewers,

We thank the reviewers for taking the time and effort to provide feedback on our submitted manuscript.

We are glad that reviewers found our proposed method sound, well-motivated, well-written and to deliver an interesting and valuable contribution to the community.

In light of the feedback, we've identified "Clarity and presentation" to be the primary area that required our attention. We have made the following improvements:

- To enhance the paper's self-containedness and technical depth, we incorporated definitions and key references that were initially omitted due to space limitations. For a comprehensive list, please refer to reviewer uLvV's reply.
- We've refined the exposition of the conditional sampling algorithm in Section 3.3, providing greater clarity to our novel contributions.
- In our experimental evaluation, we now more effectively underscore the advantages of the geometrical approach over the default case.

Lastly, we genuinely regret and apologize for the oversight regarding the preliminary version of the appendix bundled with the main PDF.

Once again, we want to thank all reviewers for your diligent review, and the overall positive evaluation of our manuscript.

---

### Decision · Program_Chairs · 2023-09-21

**Decision:**

Accept (poster)

**Comment:**

This paper proposes an algorithm for diffusion generative modelling over function spaces. Their proposed algorithm also incorporates symmetries in the model, and can be extended to manifold-valued functional models. Finally, a conditional sampling method is proposed.

The authors demonstrate clear advantage of their algorithm on a number of smaller-scale experiments, both synthetic and real-world. These provide a convincing demonstration of the value of incorporating group invariances into functional diffusion models for geometric data.

On the other hand, as noted by some reviewers, the paper's claims would be strengthened if authors can provide experimental validation on more complex and higher-dimensional problems.